# Learning under Imitative Strategic Behavior with Unforeseeable Outcomes

**Tian Xie** *xie.1379@osu.edu*
*Department of Computer Science and Engineering*
*the Ohio State University*

**Zhiqun Zuo** *zuo.167@osu.edu*
*Department of Computer Science and Engineering*
*the Ohio State University*

**Mohammad Mahdi Khalili** *khalili.17@osu.edu*
*Department of Computer Science and Engineering*
*the Ohio State University*

**Xueru Zhang** *zhang.12807@osu.edu*
*Department of Computer Science and Engineering*
*the Ohio State University*

**Reviewed on OpenReview:** *https://openreview.net/forum?id=82bNZGMNZa*

## Abstract

Machine learning systems have been widely used to make decisions about individuals who may behave strategically to receive favorable outcomes, e.g., they may genuinely *improve* the true labels or *manipulate* observable features directly to game the system without changing labels. Although both behaviors have been studied (often as two separate problems) in the literature, most works assume individuals can (i) perfectly foresee the outcomes of their behaviors when they best respond; (ii) change their features arbitrarily as long as it is affordable, and the costs they need to pay are deterministic functions of feature changes. In this paper, we consider a different setting and focus on *imitative* strategic behaviors with *unforeseeable* outcomes, i.e., individuals manipulate/improve by imitating the features of those with positive labels, but the induced feature changes are unforeseeable. We first propose a Stackelberg game to model the interplay between individuals and the decision-maker, under which we examine how the decision-maker's ability to anticipate individual behavior affects its objective function and the individual's best response. We show that the objective difference between the two can be decomposed into three interpretable terms, with each representing the decision-maker's preference for a certain behavior. By exploring the roles of each term, we theoretically illustrate how a decision-maker with adjusted preferences may simultaneously disincentivize manipulation, incentivize improvement, and promote fairness. Such theoretical results provide a guideline for decision-makers to inform better and socially responsible decisions in practice.

## 1 Introduction

Individuals subject to algorithmic decisions often adapt their behaviors strategically to the decision rule to receive a desirable outcome. As machine learning is increasingly used to make decisions about humans, there has been a growing interest to develop learning methods that explicitly consider the strategic behavior of human agents. A line of research known as *strategic classification* studies this problem, in which individuals can modify their features at costs to receive favorable predictions. Depending on whether such feature changes are to improve the actual labels genuinely (i.e., improvement) or to game the algorithms maliciously (i.e., manipulation), existing works have largely focused on learning classifiers robust against manipulation

(Hardt et al., 2016a) or designing incentive mechanisms to encourage improvement (Kleinberg & Raghavan, 2020; Bechavod et al., 2022; Xie et al., 2024). A few studies (Miller et al., 2020; Shavit et al., 2020; Horowitz & Rosenfeld, 2023) also consider the presence of both manipulation and improvement, where they exploit the causal structures of features and use *structural causal models* to capture the impacts of feature changes on labels.

To model the interplay between individuals and decision-maker, most existing works adopt (or extend based on) a *Stackelberg game* proposed by Hardt et al. (2016a), i.e., the decision-maker publishes its policy, following which individuals best respond to determine the modified features. However, these models (implicitly) rely on the following two assumptions that could make them unsuitable for certain applications: (i) individuals can perfectly foresee the outcomes of their behaviors when they best respond; (ii) individuals can change their features arbitrarily at costs, which are modeled as deterministic functions of the feature.

In other words, existing studies assume individuals know their exact feature values before and after strategic behavior. Thus, the cost can be computed precisely based on the feature changes (e.g., using functions such as $\ell_p$-norm distance). However, these may not hold in many important applications.

Consider an example of college admission, where the students' exam scores are treated as features in admission decisions. To get admitted, students may increase their scores by either cheating on exams (manipulation) or working hard (improvement). Here (i) individuals do not know the exact values of their original features (unrealized scores) and the modified features (actual scores received in an exam) when they best respond, but they have a good idea of what those score distributions would be like from their past experience; (ii) the cost of manipulation/improvement is not a function of feature change (e.g., students may cheat by hiring an imposter to take the exam and the cost of such behavior is more or less fixed). As the original feature was never realized, we cannot compute the feature change precisely and measure the cost based on it. Therefore, the existing models do not fit for these applications.

Motivated by the above (more examples are also given in App. B.2), this paper studies strategic classification with **unforeseeable outcomes**. We first propose a novel *Stackelberg game* to model the interactions between individuals and the decision-maker. Compared to most existing models (Jagadeesan et al., 2021; Levanon & Rosenfeld, 2022), ours is a *probabilistic framework* that models the outcomes and costs of strategic behavior as random variables. Indeed, this framework is inspired by the models proposed in Zhang et al. (2022); Liu et al. (2020), which only considered either strategic manipulation (Zhang et al., 2022) or improvement (Liu et al., 2020). In contrast, our model significantly extends their works by considering both manipulation and improvement behaviors, investigating agents' choices between them, and providing theoretical results and practical guideline for socially responsible decision-making in the new setting. Importantly, we focus on **imitative** strategic behavior where individuals manipulate/improve by imitating the features of those with positive labels, due to the following:

- It is inspired by imitative learning behavior in *social learning*, whereby new behaviors are acquired by copying social models' behavior. It has been well-supported by literature in psychology and social science (Bandura, 1962; 1978). Recent works (Heidari et al., 2019; Raab & Liu, 2021) in ML also model individuals' behaviors as imitating/replicating the profiles of their social models to study the impacts of fairness interventions.

- Decision-makers can detect easy-to-manipulate features (Bechavod et al., 2021) and stop using them when making decisions, so individuals can barely manipulate their features by themselves without changing labels. A better option for them is to mimic others' profiles. Such imitation-based manipulative behavior is very common in the real world (e.g., cheating, identity theft) and even becomes increasingly worrying during recent years[1].

Additionally, our model considers practical scenarios by permitting manipulation to be detected and improvement to be failed at certain probabilities, as evidenced in auditing (Estornell et al., 2021) and social

---

[1]As *COVID-19* hit the world, candidates are more commonly permitted to take exams/assessments (e.g., GRE, TOEFL, or online assessments of companies) remotely. Although many institutions are diligent in designing novel challenges to prevent candidates from directly finding the answers on the internet, the remote nature makes it easier to hire qualified imposters to take the assessments instead of them. Talha (2024) illustrated how students can let others take the GRE instead of them when the test is permitted to be taken at home.

learning (Bandura, 1962). App. A provides more related work and differences with existing models are discussed in App. B.1.

Under this model, we first study the impacts of the decision maker's ability to anticipate individual behavior. Similar to Zhang et al. (2022), we consider two types of decision-makers: non-strategic and strategic. We say a decision-maker (and its policy) is *strategic* if it has the ability to anticipate strategic behavior and accounts for this in determining the decision policies, while a *non-strategic* decision-maker ignores strategic behavior in determining its policies. Importantly, we find that the difference between the decision-maker's learning objectives under two settings can be decomposed into three interpretable terms, with each term representing the decision-maker's preference for certain behavior. By exploring the roles of each term on the decision policy and the resulting individual's best response, we further show that a strategic decision-maker with *adjusted preferences* (i.e., changing the weight of each term in the learning objective) can disincentivize manipulation while incentivizing improvement behavior.

We also consider settings where the strategic individuals come from different social groups and explore the impacts of adjusting preferences on algorithmic fairness. We show that the optimal policy under adjusted preferences may result in fairer outcomes than non-strategic policy and original strategic policy without adjustment. Moreover, such fairness promotion can be attained *simultaneously* with the goal of disincentivizing manipulation. Our contributions are summarized as follows:

1. We propose a probabilistic model to capture both improvement and manipulation; and establish a novel Stackelberg game to model the interplay between individuals and decision-maker. The individual's best response and decision-maker's (non-)strategic policies are characterized (Sec. 2).

2. We show the objective difference between non-strategic and strategic policies can be decomposed into three terms, each representing the decision-maker's preference for certain behavior (Sec. 3).

3. We study how adjusting the decision-maker's preferences can affect the optimal policy and its fairness property, as well as the resulting individual's best response (Sec. 4). We also illustrate how the decision-maker can adjust preferences to disincentivize manipulation, incentivize improvement and promote fairness in practice (Sec. 4 and App. B.5).

4. We conduct experiments on both synthetic and real data to validate the theoretical findings (Sec. 5).

## 2 Problem Formulation

Consider a group of individuals subject to some ML decisions. Each individual has an observable feature $X \in \mathbb{R}$ and a hidden label $Y \in \{0, 1\}$ indicating its qualification state ("0" being unqualified and "1" being qualified).[2] Let $\alpha := \Pr(Y = 1)$ be the population's qualification rate, and $P_{X|Y}(x|1)$, $P_{X|Y}(x|0)$ be the feature distributions of qualified and unqualified individuals, respectively. A decision-maker makes binary decisions $D \in \{0, 1\}$ ("0" being reject and "1" being accept) about individuals based on a threshold policy with acceptance threshold $\theta \in \mathbb{R}$: $\pi(x) = P_{D|X}(1|x) = \mathbf{1}(x \geq \theta)$. To receive positive decisions, individuals with information of policy $\pi$ may behave strategically by either manipulating their features or improving the actual qualifications.[3] Formally, let $M \in \{0, 1\}$ denote individual's action, with $M = 1$ being manipulation and $M = 0$ being improvement.

**Outcomes of strategic behavior.** Both manipulation and improvement result in the *shifts* of feature distribution. Specifically, for individuals who choose to **manipulate**, we assume they manipulate by "stealing" the features of those qualified (Zhang et al., 2022), e.g., students cheat on exams by hiring qualified imposters. Moreover, we assume the decision-maker can identify the manipulation behavior with probability $\epsilon \in [0, 1]$ (Estornell et al., 2021). Individuals, once getting caught manipulating, will be rejected directly. For those who decide to **improve**, they work hard to imitate the features of those qualified (Bandura, 1962; Raab

---

[2]Similar to prior work (Zhang et al., 2022; Liu et al., 2018), we present our model in one-dimensional feature space. Note that our model and results are applicable to high dimensional space, in which individuals imitate and change all features as a whole based on the joint conditional distribution $P_{X|Y}$ regardless of the dimension of $X$. The costs can be regarded as the sum of an individual's effort to change features in all dimensions.

[3]We assume all unqualified individuals who stay in the decision-making system either manipulate or improve. In other words, they already have a sufficiently large initial utility by staying in the system and trying to be qualified. We discuss the details in App. B.3.

& Liu, 2021; Heidari et al., 2019). With probability $q \in [0, 1]$, they improve the label successfully (overall $\alpha$ increases) and the features conform the distribution $P_{X|Y}(x|1)$; with probability $1 - q$, they slightly improve the features but fail to change the labels, and the improved features conform a new distribution $P^I(x)$. Throughout the paper, we make the following assumption on feature distributions.

**Assumption 2.1.** $P_{X|Y}(x|1), P_{X|Y}(x|0), P^I(x)$ *are continuous; distribution pairs* $\left(P_{X|Y}(x|1), P^I(x)\right)$ *and* $\left(P^I(x), P_{X|Y}(x|0)\right)$ *satisfy the strict monotone likelihood ratio property, i.e.* $\frac{P^I(x)}{P_{X|Y}(x|0)}$ *and* $\frac{P_{X|Y}(x|1)}{P^I(x)}$ *are increasing in* $x \in \mathbb{R}$.

Assumption 2.1 is relatively mild and has been widely used (e.g., (Tsirtsis et al., 2019; Zhang et al., 2020b)). It can be satisfied by a wide range of distributions (e.g., exponential, Gaussian) and the real data (e.g., FICO data used in Sec. 5). It implies that an individual is more likely to be qualified as feature value increases. Meanwhile, compared to the unqualified individuals, the individuals who improve but fail also tend to have higher feature values. Individuals have a good knowledge of their true qualifications by observing their peers or previous individuals who received positive decisions Raab & Liu (2021), and only unqualified individuals have incentives to take action Dong et al. (2018) since $P_{X|Y}(x|1)$ is always the best attainable outcome (as manipulation and improvement only bring additional cost but no benefit to qualified individuals).

## 2.1 Individual's best response.

An individual incurs a random cost $C_M \geq 0$ when manipulating the features (Zhang et al., 2022), while incurring a random cost $C_I \geq 0$ when improving the qualifications (Liu et al., 2020). The realizations of these random costs are known to individuals when determining their action $M$; while the decision-maker only knows the cost distributions. Thus, the best response that the decision-maker expects from individuals is the probability of manipulation/improvement. Figure 1 illustrates the strategic interaction between them.

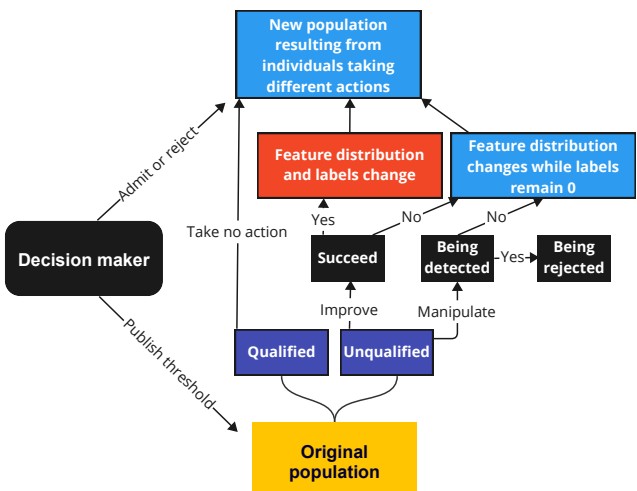

Figure 1: Illustration of the strategic interaction

Formally, given a policy $\pi(x) = \mathbf{1}(x \geq \theta)$ with threshold $\theta$, an individual chooses to manipulate only if the expected utility attained under manipulation $U_M(\theta)$ outweighs the utility under improvement $U_I(\theta)$. Suppose an individual benefits $w = 1$ from the acceptance, and 0 from the rejection. Given that each individual only knows his/her label $y \in \{0, 1\}$ and the conditional feature distributions $P_{X|Y}$ but **not** the exact values of the feature $x$, the expected utilities $U_M(\theta)$ and $U_I(\theta)$ can be computed as the expected benefit minus the cost of action, as given below.

$$U_M(\theta) = F_{X|Y}(\theta|0) - F_{X|Y}(\theta|1) - \epsilon(1 - F_{X|Y}(\theta|1)) - C_M$$

$$U_I(\theta) = F_{X|Y}(\theta|0) - q \cdot F_{X|Y}(\theta|1) - (1 - q) \cdot F^I(\theta) - C_I$$

where $F_{X|Y}(x|1)$, $F_{X|Y}(x|0)$, $F^I(x)$ are cumulative density function (CDF) of $P_{X|Y}(x|1)$, $P_{X|Y}(x|0)$, $P^I(x)$, respectively. Given the threshold $\theta$, the decision-maker can anticipate the probability that an unqualified individual chooses to manipulate as $P_M(\theta) = \Pr(U_M(\theta) > U_I(\theta))$, which can further be written as follows (derivations and more explanation details in App. D.1):

$$P_M(\theta) = \Pr\left((1-q) \cdot \left(F^I(\theta) - F_{X|Y}(\theta|1)\right) - \epsilon\left(1 - F_{X|Y}(\theta|1)\right) \geq C_M - C_I\right) \tag{1}$$

The above formulation captures the imitative strategic behavior with unforeseeable outcomes (e.g., college admission example in Sec. 1): individuals best respond based on feature distributions but not the realizations, and the imitation costs (e.g., hiring an imposter) for individuals from the same group follow the same distribution (Liu et al., 2020), as opposed to being a function of feature changes. equation 1 above can further be written based on CDF of $C_M - C_I$, i.e., the difference between manipulation and improvement costs. We make the following assumption on its PDF.

**Assumption 2.2.** *The PDF $P_{C_M-C_I}(x)$ is continuous with $P_{C_M-C_I}(x) > 0$ for $x \in (-\epsilon, 1-q)$.*

Assumption 2.2 is mild only to ensure the manipulation is possible under all thresholds $\theta$. Under the Assumption, we can study the impact of acceptance threshold $\theta$ on manipulation probability $P_M(\theta)$.

**Theorem 2.3** (Manipulation Probability). *Under Assumption 2.2, $P_M(\theta)$ is continuous and satisfies the following: (i) If $q + \epsilon \geq 1$, then $P_M(\theta)$ strictly increases. (ii) If $q + \epsilon < 1$, then $P_M(\theta)$ first increases and then decreases, thereby existing a unique maximizer $\theta_{max}$. Moreover, maximizer $\theta_{max}$ increases in $q$ and $\epsilon$.*

Thm. 2.3 shows that an individual's best response highly depends on the success rate of improvement $q$ and the identification rate of manipulation $\epsilon$. When $q + \epsilon \geq 1$ (i.e., improvement can succeed or/and manipulation is detected with high probability), individuals are more likely to manipulate as $\theta$ increases. Note that although individuals are generally more likely to benefit from improvement than manipulation, as $\theta$ increases to the maximum (i.e., when the decision-maker barely admits anyone), the "net benefit" of improvement compared to manipulation will finally diminish to 0 because both actions are useless. Thus, more individuals tend to manipulate under larger $\theta$, making $P_M(\theta)$ strictly increasing and reaching the maximum. When $q + \epsilon < 1$, more individuals are incentivized to improve as the threshold gets farther away from $\theta_{max}$. This is because the manipulation in this case incurs a higher benefit than improvement at $\theta_{max}$. As the threshold increases/decreases from $\theta_{max}$ to the minimum/maximum (i.e., the decision-maker either admits almost everyone or no one), the "net benefit" of manipulation compared to improvement decreases to 0 or $-\epsilon$. Thus, $P_M(\theta)$ decreases as $\theta$ increases/decreases from $\theta_{max}$.

## 2.2 Decision-maker's optimal policy

Suppose the decision-maker receives benefit $u$ (resp. penalty $-u$) when accepting a qualified (resp. unqualified) individual, then the decision-maker aims to find an optimal policy that maximizes its expected utility $\mathbb{E}[R(D,Y)]$, where utility is $R(1,1) = u, R(1,0) = -u, R(0,1) = R(0,0) = 0$.

As mentioned in Sec. 1, we consider *strategic* and *non-strategic* decision makers. Because the former can anticipate individual's strategic behavior while the latter cannot, their learning objectives $\mathbb{E}[R(D,Y)]$ are different. As a result, their respective optimal policies are also different.

**Non-strategic optimal policy.** Without accounting for strategic behavior, the non-strategic decision-maker's learning objective $\widehat{U}(\pi)$ under policy $\pi$ is given by:

$$\widehat{U}(\pi) = \int_X \{u\alpha P_{X|Y}(x|1) - u(1-\alpha)P_{X|Y}(x|0)\}\pi(x)\,dx \tag{2}$$

Under Assumption 2.1, it has been shown in Zhang et al. (2020b) that the optimal non-strategic policy that maximizes $\widehat{U}(\pi)$ is a threshold policy with threshold $\widehat{\theta}^*$ satisfying $\frac{P_{X|Y}(\widehat{\theta}^*|1)}{P_{X|Y}(\widehat{\theta}^*|0)} = \frac{1-\alpha}{\alpha}$.

**Strategic optimal policy.** Given cost and feature distributions, a strategic decision-maker can anticipate an individual's best response (equation 1) and incorporate it in determining its optimal policy. Under a

threshold policy $\pi(x) = \mathbf{1}(x \geq \theta)$, the objective $U(\pi)$ can be written as a function of $\theta$, i.e.,

$$
\begin{aligned}
U(\theta) = & u\Big(\alpha + (1-\alpha)(1 - P_M(\theta))q\Big) \cdot \big(1 - F_{X|Y}(\theta|1)\big) - u(1-\alpha)\Big((1-\epsilon) \cdot P_M(\theta) \cdot \big(1 - F_{X|Y}(\theta|1)\big) \\
& + (1 - P_M(\theta)) \cdot (1-q)(1 - F^I(\theta))\Big)
\end{aligned}
\tag{3}
$$

The policy that maximizes the above objective function $U(\theta)$ is the strategic optimal policy. We denote the corresponding optimal threshold as $\theta^*$. Compared to non-strategic policy, $U(\theta)$ also depends on $q, \epsilon, P_M(\theta)$ and is rather complicated. Nonetheless, we will show in Sec. 3 that $U(\theta)$ can be justified and decomposed into several interpretable terms.

## 3 Decomposition of the Objective Difference

In Sec. 2.2, we derived the learning objective functions of both strategic and non-strategic decision-makers (expected utilities $U$ and $\widehat{U}$). Next, we explore how the individual's choice of improvement or manipulation affects decision-maker's utility. Define $\Phi(\theta) = U(\theta) - \widehat{U}(\theta)$ as the *objective difference* between strategic and non-strategic decision-makers, we have:

$$
\Phi(\theta) = u(1-\alpha) \cdot \Big(\phi_1(\theta) - \phi_2(\theta) - \phi_3(\theta)\Big)
\tag{4}
$$

where

$$
\begin{aligned}
\phi_1(\theta) &= \big(1 - P_M(\theta)\big) \cdot q \cdot \big(1 - F_{X|Y}(\theta|0) + 1 - F_{X|Y}(\theta|1)\big) \\
\phi_2(\theta) &= \big(1 - P_M(\theta)\big) \cdot (1-q) \cdot \big(F_{X|Y}(\theta|0) - F^I(\theta)\big) \\
\phi_3(\theta) &= P_M(\theta)\big((1-\epsilon)\big(1 - F_{X|Y}(\theta|1)\big) - \big(1 - F_{X|Y}(\theta|0)\big)\big)
\end{aligned}
$$

As shown in equation 4, the objective difference $\Phi$ can be decomposed into three terms $\phi_1, \phi_2, \phi_3$. It turns out that each term is interpretable and indicates the impact of a certain type of individual behavior on the decision-maker's utility. We discuss these in detail as follows.

1. **Benefit from the successful improvement** $\phi_1$: additional *benefit* the decision-maker gains due to the successful improvement of individuals (as the successful improvement causes label change). In the college admission example, $\phi_1$ stands for the utility gain caused by students studying hard to be qualified.

2. **Loss from the failed improvement** $\phi_2$: additional *loss* the decision-maker suffers due to the individuals' failure to improve; this occurs because individuals who fail to improve only experience feature distribution shifts from $P_{X|Y}(x|0)$ to $P^I(x)$ but labels remain. In the college admission example, $\phi_2$ stands for the utility loss by students who seem to try but fail to be qualified.

3. **Loss from the manipulation** $\phi_3$: additional *loss* the decision-maker suffers due to the successful manipulation of individuals; this occurs because individuals who manipulate successfully only change $P_{X|Y}(x|0)$ to $P_{X|Y}(x|1)$ but the labels remain unqualified. In the college admission example, $\phi_3$ stands for the utility loss caused by students who cheat/hire imposters.

Note that in Zhang et al. (2022), the objective difference $\Phi(\theta)$ has only one term corresponding to the additional loss caused by strategic manipulation. Because our model further considers improvement behavior, the impact of an individual's strategic behavior on the decision-maker's utility gets more complicated. We have illustrated above that in addition to the loss from manipulation $\phi_3$, the improvement behavior also affects decision-maker's utility. Importantly, such an effect can be either positive (if the improvement is successful) or negative (if the improvement fails).

The decomposition of the objective difference $\Phi(\theta)$ highlights the connections between three types of policies: 1) non-strategic policy without considering individual's behavior; 2) strategic policy studied in Zhang et al. (2022) that only considers manipulation, 3) strategic policy studied in this paper that considers both manipulation and improvement. Specifically, by removing $\phi_1, \phi_2, \phi_3$ (resp. $\phi_1, \phi_2$) from the objective function $U(\theta)$, the strategic policy studied in this paper would reduce to the non-strategic policy (resp. strategic policy studied in Zhang et al. (2022)). Based on this observation, we regard $\phi_1, \phi_2, \phi_3$ each as the decision-maker's **preference** to a certain type of individual behavior, and define a general strategic decision-maker with adjusted preferences.

### 3.1 Strategic decision-maker with adjusted preferences

We consider general strategic decision-makers who find the optimal decision policy by maximizing $\widehat{U}(\theta) + \Phi(\theta, k_1, k_2, k_3)$ with

$$\Phi(\theta, k_1, k_2, k_3) = k_1 \cdot \phi_1(\theta) - k_2 \cdot \phi_2(\theta) - k_3 \cdot \phi_3(\theta) \tag{5}$$

where $k_1, k_2, k_3 \geq 0$ are weight parameters; different combinations of weights correspond to different preferences of the decision-maker. We give some examples below:

1. **Original strategic decision-maker:** the one with $k_1 = k_2 = k_3 = u(1 - \alpha)$ whose learning objective function $U$ follows equation 3; it considers both improvement and manipulation.

2. **Improvement-encouraging decision-maker:** the one with $k_1 > 0$ and $k_2 = k_3 = 0$; it only considers strategic improvement and only values the improvement benefit while ignoring the loss caused by the failure of improvement.

3. **Manipulation-proof decision-maker:** the one with $k_3 > 0$ and $k_1 = k_2 = 0$; it is only concerned with strategic manipulation, and the goal is to prevent manipulation.

4. **Improvement-proof decision-maker:** the one with $k_2 > 0$ and $k_1 = k_3 = 0$; it only considers improvement but the goal is to avoid loss caused by the failed improvement.

The above examples show that a decision-maker, by changing the weights $k_1, k_2, k_3$ could find a policy that encourages certain types of individual behavior (as compared to the original policy $\theta^*$). Although the decision-maker can impact an individual's behavior by adjusting its preferences via $k_1, k_2, k_3$, we emphasize that the **actual utility** it receives from the strategic individuals is always determined by $U(\theta)$ given in equation 3. Indeed, we can regard the framework with adjusted weights (equation 5) as a *regularization* method. We discuss this in more detail in App. B.4.

## 4 Impacts of Adjusting Preferences

Next, we investigate the impacts of adjusting preferences. We aim to understand how a decision-maker can adjust preferences (i.e., changing $k_1, k_2, k_3$) to affect the optimal policy (Sec. 4.1) and its fairness property (Sec. 4.3), as well as the resulting individual's best response (Sec. 4.2).

### 4.1 Preferences shift the optimal threshold

We will start with the original strategic decision-maker (with $k_1 = k_2 = k_3 = u(1 - \alpha)$) whose objective function follows equation 3, and then investigate how adjusting preferences could affect the decision-maker's optimal policy.

**Complex nature of original strategic decision-maker.** Unlike the non-strategic optimal policy, the analytical solution of strategic optimal policy that maximizes equation 3 is not easy to find. In fact, the utility function $U(\theta)$ of the original strategic decision-maker is highly complex, and the optimal strategic threshold $\theta^*$ may change significantly as $\alpha, F_{X|Y}, F^I, C_M, C_I, \epsilon, q$ vary. In App. C.2, we demonstrate the complexity of $U(\theta)$, which may change drastically as $\alpha, \epsilon, q$ vary. Although we cannot find the strategic optimal threshold precisely, we may still explore the impacts of the decision-maker's anticipation of strategic behavior on its policy (by comparing the strategic threshold $\theta^*$ with the non-strategic $\widehat{\theta}^*$), as stated in Thm. 4.1 below.

**Theorem 4.1** (Comparison of strategic and non-strategic policy)**.** *If* $\min_\theta P_M(\theta) \leq 0.5$*, then there exists* $\widehat{q} \in (0, 1)$ *such that* $\forall q \geq \widehat{q}$*, the strategic optimal* $\theta^*$ *is always lower than the non-strategic* $\widehat{\theta}^*$*.*

Thm. 4.1 identifies a condition under which the strategic policy over-accepts individuals compared to the non-strategic one. Specifically, $\min_\theta P_M(\theta) \leq 0.5$ ensures that there exist policies under which the majority of individuals prefer improvement over manipulation. Intuitively, under this condition, a strategic decision-maker by lowering the threshold (from $\widehat{\theta}^*$) may encourage more individuals to improve. Because $q$ is sufficiently large, more improvement brings more benefit to the decision-maker.

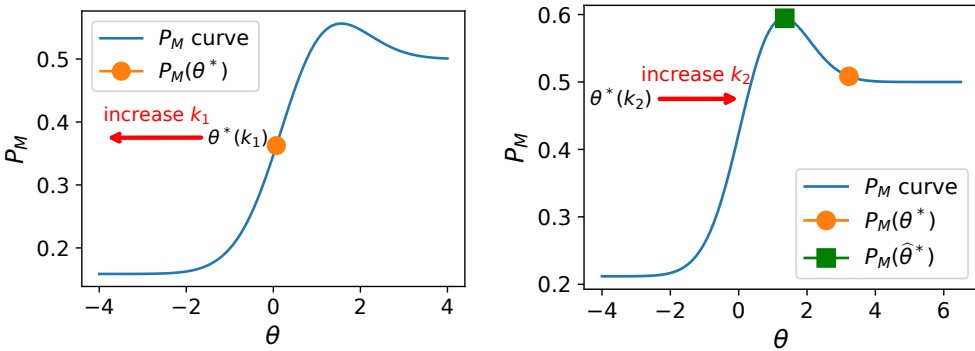

Figure 2: Illustration of scenario 1 (left) and scenario 2 (right) in Thm. 4.5: adjusting preferences decreases manipulation probability $P_M(\theta)$.

**Optimal threshold under adjusted preferences.** Despite the intricate nature of $U(\theta)$, the optimal strategic threshold may be *shifted* by adjusting the decision-maker's *preferences*, i.e. changing the weights $k_1, k_2, k_3$ assigned to $\phi_1, \phi_2, \phi_3$ in equation 5. Next, we examine how the optimal threshold can be affected compared to the original strategic threshold by adjusting the decision-maker's preferences. Denote $\theta^*(k_i)$ as the strategic optimal threshold attained by adjusting weight $k_i, i \in \{1, 2, 3\}$ of the original objective function $U(\theta)$. The results are summarized in Table 1. Specifically, the threshold gets lower as $k_1$ increases (Prop. 4.2). Adjusting $k_2$ or $k_3$ may result in the optimal threshold moving toward both directions, but we can identify sufficient conditions when adjusting $k_2$ or $k_3$ pushes the optimal threshold to move toward one direction (Prop. 4.3 and 4.4).

Table 1: The impact of adjusted preferences on $\theta^*(k_i)$ compared to the original strategic threshold $\theta^*$.

| Adjusted weight | Preference | Threshold shift |
|---|---|---|
| Increase $k_1$ | Encourage improvement | $\theta^*(k_1) < \theta^*$ |
| Increase $k_2$ | Discourage improvement | $\theta^*(k_2) \lessgtr \theta^*$ |
| Increase $k_3$ | Discourage manipulation | $\theta^*(k_3) \lessgtr \theta^*$ |

**Proposition 4.2.** *Increasing $k_1$ results in a lower optimal threshold $\theta^*(k_1) < \theta^*$. Moreover, when $k_1$ is sufficiently large, $\theta^*(k_1) < \widehat{\theta}^*$.*

**Proposition 4.3.** *When $\alpha \leq 0.5$ (the majority of the population is unqualified), increasing $k_2$ results in a higher optimal threshold $\theta^*(k_2) > \theta^*$. Moreover, when $k_2$ is sufficiently large, $\theta^*(k_2) > \widehat{\theta}^*$.*

**Proposition 4.4.** *For any feature distribution $P_{X|Y}$, there exists an $\bar{\epsilon} \in (0, 1)$ such that whenever $\epsilon \geq \bar{\epsilon}$, increasing $k_3$ results in a lower optimal threshold $\theta^*(k_3) < \theta^*$.*

Prop. 4.2 to 4.4 reveal that adjusting preferences may lead to predictable changes of optimal strategic thresholds under certain conditions. So far we have shown how the optimal threshold can be shifted as the decision maker's preferences change. Next, we explore the impacts of threshold shifts on individuals' behaviors and show how a decision-maker with adjusted preferences can (dis)incentivize manipulation and influence fairness.

### 4.2 Preferences as (dis)incentives for manipulation

In Thm. 2.3, we explored the impacts of threshold $\theta$ on individuals' best responses $P_M(\theta)$. Combined with our knowledge of the relationship between adjusted preferences and policy (Sec. 4.1), we can further analyze how adjusting preferences affect individuals' responses. Next, we illustrate how a decision-maker may disincentivize manipulation (or equivalently, incentivize improvement) by adjusting its preferences.

**Theorem 4.5** (Preferences serve as (dis)incentives). *Compared to the original strategic policy $\theta^*$, decision-makers can adjust preferences to disincentivize manipulation (i.e., $P_M(\theta)$ decreases) under certain scenarios. Specifically,*

1. When **either** of the following is satisfied, and the decision-maker adjusts preferences by increasing $k_1$:

$$(i).\ q + \epsilon \geq 1; \qquad\qquad (ii).\ \frac{P_{X|Y}(\theta^*|1)}{P^I(\theta^*)} \leq \frac{1-q}{1-q-\epsilon}.$$

2. When **both** of the followings are satisfied, and the decision-maker adjusts preferences by increasing $k_2$:

$$(i).\ q + \epsilon < 1 \text{ and } \alpha < 0.5; \qquad (ii).\ \frac{P_{X|Y}(\theta^*|1)}{P^I(\theta^*)} > \frac{1-q}{1-q-\epsilon} \text{ and } P_M(\widehat{\theta^*}) > F_{C_M - C_I}(0).$$

*Moreover, when $k_1$ (for scenario 1) or $k_2$ (for scenario 2) are sufficiently large, adjusting preferences also disincentivize the manipulation compared to the non-strategic policy $\widehat{\theta}^*$.*

Thm. 4.5 identifies conditions under which a decision-maker can disincentivize manipulation directly by adjusting its preferences. The condition $q + \epsilon \lessgtr 1$ determines whether the best response $P_M(\theta)$ is strictly increasing or single-peaked (Thm. 2.3); the condition $\frac{P_{X|Y}(\theta^*|1)}{P^I(\theta^*)} \lessgtr \frac{1-q}{1-q-\epsilon}$ implies that $\theta^*$ is lower/higher than $\theta_{max}$ in Thm. 2.3. In Fig. 2, we illustrate Thm. 4.5 where the left (resp. right) plot corresponds to scenario 1 (resp. scenario 2). Because increasing $k_1$ (resp. $k_2$) results in a lower (resp. higher) threshold than $\theta^*$, the resulting manipulation probability $P_M$ is lower for both scenarios. The detailed experimental setup and more illustrations are in App. C.

### 4.3 Preferences shape algorithmic fairness

The threshold shifts under adjusted preferences further allow us to compare these policies against a certain fairness measure. In this section, we consider strategic individuals from two social groups $\mathcal{G}_a, \mathcal{G}_b$ distinguished by some protected attribute $S \in \{a, b\}$ (e.g., race, gender). Similar to Zhang et al. (2019; 2020b; 2022), we assume the protected attributes are observable and the decision-maker uses *group-dependent* threshold policy $\pi_s(x) = \mathbf{1}(x \geq \theta_s)$ to make decisions about $\mathcal{G}_s, s \in \{a, b\}$. The optimal threshold for each group can be found by maximizing the utility associated with that group: $\max_{\theta_s} \mathbb{E}[R(D, Y)|S = s]$.

**Fairness measure.** We consider a class of group fairness notions that can be represented in the following form (Zhang et al., 2020a; Zhang & Liu, 2021):

$$\mathbb{E}_{X \sim P_a^{\mathcal{C}}}[\pi_a(X)] = \mathbb{E}_{X \sim P_b^{\mathcal{C}}}[\pi_b(X)]$$

where $P_s^{\mathcal{C}}$ is some probability distribution over $X$ associated with fairness metric $\mathcal{C}$. For instance, under equal opportunity (`EqOpt`) fairness (Hardt et al., 2016b), $P_s^{\texttt{EqOpt}}(x) = P_{X|YS}(x|1, s)$; under demographic parity (`DP`) fairness (Barocas et al., 2019), $P_s^{\texttt{DP}}(x) = P_{X|S}(x|s)$ .

For threshold policy with thresholds $(\theta_a, \theta_b)$, we measure the unfairness as $\big|\mathbb{E}_{X \sim P_a^{\mathcal{C}}}[\mathbf{1}(x \geq \theta_a)] - \mathbb{E}_{X \sim P_b^{\mathcal{C}}}[\mathbf{1}(x \geq \theta_b)]\big|$. Define the *advantaged group* as the group with larger $\mathbb{E}_{X \sim P_s^{\mathcal{C}}}[\mathbf{1}(X \geq \widehat{\theta}_s^*)]$ under non-strategic optimal policy $\widehat{\theta}_s^*$, i.e., the group with the larger true positive rate (resp. positive rate) under `EqOpt` (resp. `DP`) fairness, and the other group as *disadvantaged group*.

**Mitigate unfairness with adjusted preferences.** Next, we compare the unfairness of different policies and illustrate that decision-makers with adjusted preferences may result in fairer outcomes, as compared to both the original strategic and the non-strategic policy.

**Theorem 4.6** (Promote fairness while disincentivizing manipulation)**.** *Without loss of generality, let $\mathcal{G}_a$ be the advantaged group and $\mathcal{G}_b$ disadvantaged. A strategic decision-maker can always simultaneously disincentivize manipulation and promote fairness in any of the following scenarios:*

1. *When condition 1.(i) **or** 1.(ii) in Thm. 4.5 holds for both groups, and the decision-maker adjusts the preferences by increasing $k_1$ for both groups.*

2. *When condition 2.(i) **and** 2.(ii) in Thm. 4.5 hold for both groups and the decision-maker adjusts the preferences by increasing $k_2$ for both groups.*

3. *When condition 1.(i) or 1.(ii) holds for $\mathcal{G}_a$, condition 2.(i) and 2.(ii) hold for $\mathcal{G}_b$, and the decision-maker adjusts preferences by increasing $k_1$ for $\mathcal{G}_a$ and $k_2$ for $\mathcal{G}_b$.*

**Corollary 4.7.** *If none of the three scenarios in Thm. 4.6 holds, adjusting preferences is not guaranteed to promote fairness and disincentivize manipulation simultaneously.*

Thm. 4.6 identifies *all* scenarios under which a decision-maker can simultaneously promote fairness and disincentivize manipulation by simply adjusting $k_1, k_2$. Otherwise, it is not guaranteed that both objectives can be achieved at the same time, as stated in Corollary 4.7.

**A practical guideline for socially responsible decisions.** The theoretical results in Thm. 4.5 give conditions under which the decision-maker can adjust preferences to disincentivize manipulation and encourage improvement, while the ones in Thm. 4.6 further sheds light on how the decision-maker can make explainable and socially responsible decisions under the unforeseeable strategic individual behavior: instead of adding separate regularizers to prevent manipulation or promote fairness, we show that the decision-maker may adjust their preferences in an interpretable way to disincentivize manipulation, incentivize improvement and promote fairness at the same time.

However, applying our theoretical results in practice needs access to several model parameters such as $q, \epsilon, C_M, C_I, F^I$ which can be estimated empirically. In App. B.5, we assume the decision-maker needs to estimate all parameters except the feature distribution for the qualified individuals $P_{X|Y}(x|1)$ and the qualification rate $\alpha$. Then we provide an ***estimation procedure*** for the parameters using controlled experiments on an experimental population[4]. With the model parameters, the decision-maker can derive accurate expressions for the decompositions $\phi_1, \phi_2, \phi_3$ and verify whether any condition in Thm. 4.5 and 4.6 holds. This enables the decision-maker to design a policy by adjusting preferences. For example, if estimated probabilities $q + \epsilon \geq 1$, then the decision-maker may train a policy with increased $k_1$ to disincentivize manipulation behavior (by Thm. 4.5, case 1).

## 5 Experiments

We conduct experiments on both synthetic Gaussian data and FICO score data (Hardt et al., 2016b)[5].

**FICO data (Hardt et al., 2016b).** FICO scores are widely used in the US to predict people's credit worthiness. We use the preprocessed dataset containing the CDF of scores $F_{X|S}(x|s)$, qualification likelihoods $P_{Y|XS}(1|x,s)$, and qualification rates $\alpha_s$ for four racial groups (Caucasian, African American, Hispanic, Asian). All scores are normalized to $[0,1]$. Similar to Zhang et al. (2022), we use these to estimate the conditional feature distributions $P_{X|YS}(x|y,s)$ using beta distribution $Beta(a_{ys}, b_{ys})$. The results are shown in Fig. 9. We assume the improved feature distribution $P^I(x) \sim Beta\left(\frac{a_{1s}+a_{0s}}{2}, \frac{b_{1s}+b_{0s}}{2}\right)$ and $C_M - C_I \sim \mathcal{N}(0, 0.25)$ for all groups, under which Assumption 2.2 and 2.1 are satisfied (see Fig. 8). We also considered other feature/cost distributions and observed similar results. Note that for each group $s$, the decision-maker finds its own optimal threshold $\left(\theta_s^* \text{ or } \theta_s^*(k_i) \text{ or } \widehat{\theta}_s^*\right)$ by maximizing the utility associated with that group, i.e., $\max_{\theta_s} \mathbb{E}[R(D,Y)|S=s]$.

We first examine the impact of the decision-maker's anticipation of strategic behavior on policies. In Fig. 21 (App. C.1), the strategic $\theta_s^*$ and non-strategic optimal threshold $\widehat{\theta}_s^*$ are compared for each group under different $q$ and $\epsilon$. The results are consistent with Thm. 4.1, i.e., under certain conditions, $\theta_s^*$ is lower than $\widehat{\theta}_s^*$ when $q$ is sufficiently large.

We also examine the individual best responses. Fig. 3 shows the manipulation probability $P_M(\theta)$ as a function of threshold $\theta$ for Caucasians and Asians (blue) versus African Americans and Hispanic (orange) when $q = 0.3, \epsilon = 0.5$. For all groups, there exists a unique $\theta_{max}$ that maximizes the manipulation probability. These are consistent with Thm. 2.3. We also indicate the manipulation probabilities under original strategic optimal thresholds $\theta_s^*$; The results indicate that original strategic optimal thresholds may cause the disadvantaged groups to have higher manipulation probabilities.

Note that the scenario considered in Fig. 3 satisfies the condition *1.(ii)* in Thm. 4.5, because the original strategic $\theta_s^* < \theta_{max}$ for both groups. We further conduct experiments in this setting to evaluate the impacts of adjusted preferences. We first adopt `EqOpt` as the fairness metric, under which $\mathbb{E}_{X \sim P_s^C}[\mathbf{1}(X \geq \widehat{\theta})] =$

---

[4]Miller et al. (2020) already argued that designing policy for any strategic classification problem is a non-trivial causal modeling problem, thereby making the controlled experiments necessary for applications in practice.

[5]code available at https://github.com/osu-srml/Unforeseeable-SC/tree/main

Table 2: Comparison between three types of optimal thresholds (FICO data). For Utility and $P_M$ values, the left value in parenthesis is associated with the advantaged group (Caucasian, Asian), while the right is for the disadvantaged group (African American, Hispanic). The first tabular is for Caucasian/African American while the second is for Asian/Hispanic.

| Threshold | Utility | $P_M$ | Unfairness (EqOpt) |
|---|---|---|---|
| Non-strategic | $(0.698, 0.171)$ | $(0.331, 0.513)$ | 0.136 |
| Original strategic | $(0.704, 0.203)$ | $(0.211, 0.278)$ | 0.055 |
| Adjusted strategic | $(0.701, 0.189)$ | $(0.140, 0.155)$ | 0.028 |

| Threshold category | Utility | $P_M$ | Unfairness (EqOpt) |
|---|---|---|---|
| Non-strategic | $(0.726, 0.427)$ | $(0.115, 0.322)$ | 0.089 |
| Original strategic | $(0.734, 0.448)$ | $(0.055, 0.161)$ | 0.047 |
| Adjusted strategic | $(0.726, 0.434)$ | $(0.023, 0.070)$ | 0.022 |

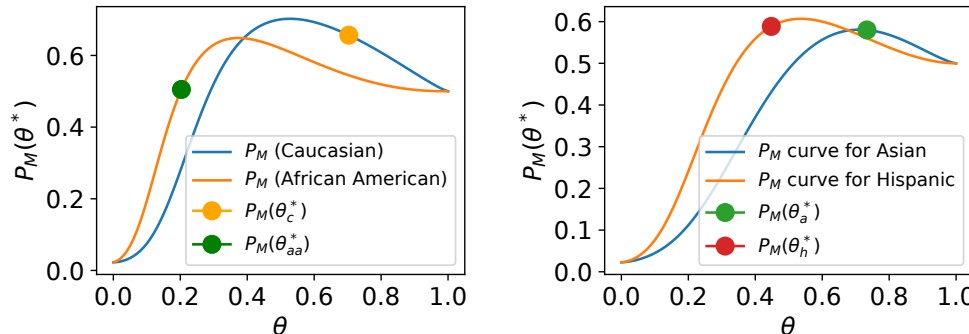

Figure 3: $P_M(\theta)$ of Caucasian and African American (left plot) and of Asian and Hispanic (right plot).

$F_{X|YS}(\theta|1, s)$ and the unfairness measure of group $\mathcal{G}_a, \mathcal{G}_b$ can be reduced to $\left| F_{X|YS}(\theta|1, a) - F_{X|YS}(\theta|1, b) \right|$. Experiments for other fairness metrics are in App. C.1. The results are shown in Fig. 4, where dashed red and dashed blue curves are manipulation probabilities under non-strategic $\widehat{\theta}^*$ and strategic $\theta^*(k_1)$, respectively. Solid red and solid blue curves are the actual utilities $U(\widehat{\theta}^*)$ and $U(\theta^*(k_1))$ received by the decision-maker. The difference between the two dotted green curves measures the unfairness between Caucasians/African Americans or Asians/Hispanics. All weights are normalized such that $k_1 = 1$ corresponds to the original strategic policy, and $k_1 > 1$ indicates the policies with adjusted preferences. Results show that when condition *1(ii)* in Thm. 4.5 is satisfied, increasing $k_1$ can simultaneously disincentivize manipulation ($P_M$ decreases with $k_1$) and improve fairness. These validate Thm. 4.5 and 4.6.

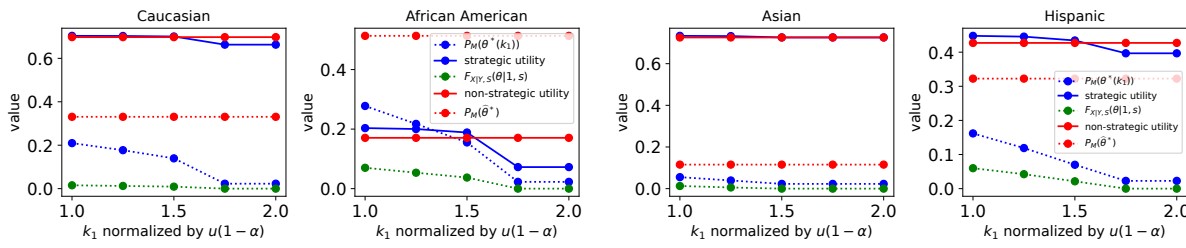

Figure 4: Impact of adjusted preferences: Caucasian and African American (left plot), Asian and Hispanic (right plot)

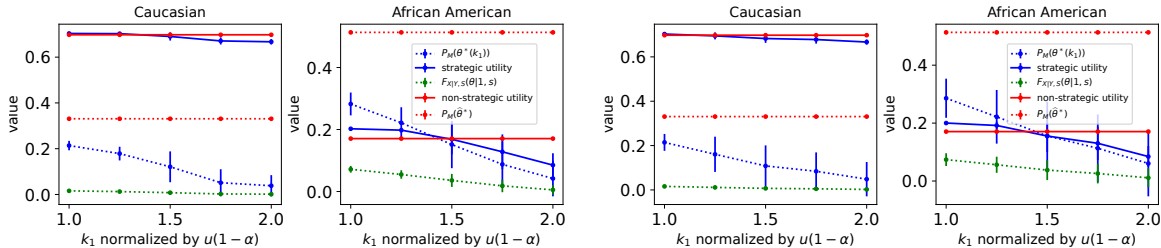

Figure 5: Impact of adjusted preferences (FICO data) when there is a Gaussian noise on $q$. The noises have 0 mean, and $0.05, 0.1$ standard deviation from the left two plots to the right two plots.

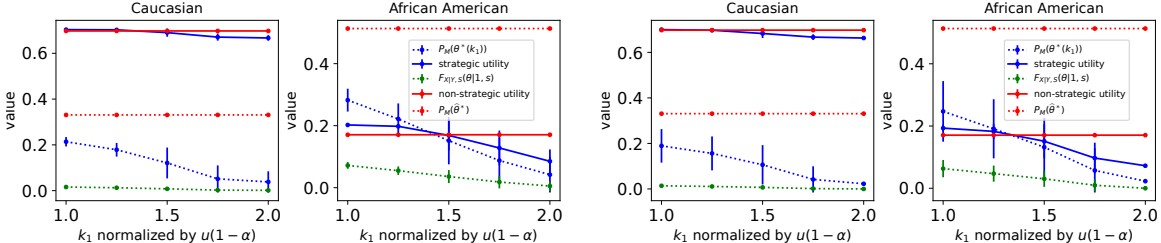

Figure 6: Impact of adjusted preferences (FICO data) when there is a Gaussian noise on $\epsilon$. The noises have 0 mean, and $0.05, 0.1$ standard deviation from the left two plots to the right two plots.

Tab. 2 compares the non-strategic $\widehat{\theta}^*$, original strategic $\theta^*$, and adjusted strategic $\theta^*(k_1)$ when $k_{1,c} = k_{1,aa} = 1.5$. It shows that decision-makers by adjusting preferences can significantly mitigate unfairness and disincentivize manipulation, with only slight decreases in utilities.

**Robustness of results when $q, \epsilon$ are noisy.** We also present experiments to relax the assumption that the decision-maker knows $q, \epsilon$ exactly on the Caucasian/African groups. Instead, they only know $q + \delta$ or $\epsilon + \delta$ where $\delta$ is a Gaussian noise. We do 10 rounds of simulations and produce plots with expectation and error bars similar to Fig. 4 (Fig. 5 shows the results with noisy $q$, while Fig. 6 shows the results with noisy $\epsilon$). The results show adjusting $k$ still works under noisy $q$ and $\epsilon$.

**Gaussian Data.** We also validate our theorems on synthetic data with Gaussian distributed $P_{X|YS}$ in App. C.2. Specifically, we examined the impacts of adjusting preferences on decision policies, individual's best response, and algorithmic fairness. As shown in Fig. 19, 20 and Tab. 5, 6, 7, these results are consistent with theorems, i.e., adjusting preferences can effectively disincentivize manipulation and improve fairness. Notably, we considered all three scenarios in Thm. 4.5 when condition *1.(i)* or *1.(ii)* or *2* is satisfied. For each scenario, we illustrate the individual's best response $P_M$ in Fig. 19 and show that manipulation can be disincentivized by adjusting preferences, i.e., increasing $k_1$ under condition *1.(i)* or *1.(ii)*, or increasing $k_2$ under condition *2*.

## 6 Conclusions & Limitations

This paper proposes a novel probabilistic framework and formulates a Stackelberg game to tackle imitative strategic behavior with unforeseeable outcomes. Moreover, the paper provides an interpretable decomposition for the decision-maker to incentivize improvement and promote fairness simultaneously. The theoretical results depend on some (mild) assumptions and are subject to change when $\epsilon, q, C_M, C_I$ change. Although we provide a practical estimation procedure to estimate the model parameters, it still remains a challenge to estimate model parameters accurately due to the expensive nature of doing controlled experiments. This may bring uncertainties in applying our framework accurately in real applications.

## Broader Impact Statement

We believe our proposed framework can promote socially responsible machine learning under strategic classification settings since the outcomes of strategic behaviors in many real-world settings are imitative and unforeseeable, thereby being more appropriately captured by our model. However, as mentioned in Sec. 6, we need certain assumptions and an estimation procedure to apply the model in practice, which may bring unexpected social outcomes.

## Acknowledgement

This material is based upon work supported by the U.S. National Science Foundation under award IIS-2202699 and IIS-2416895, by OSU President's Research Excellence Accelerator Grant, and grants from the Ohio State University's Translational Data Analytics Institute and College of Engineering Strategic Research Initiative.

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

## A Related Work

### A.1 Strategic classification

Generally, strategic behaviors can cause feature and label distribution of individuals to shift, which have long been closely related to concept drift (Lu et al., 2018), preference shift (Carroll et al., 2022), and algorithm recourse (Karimi et al., 2022). Strategic classification has been extensively studied since (Hardt et al., 2016a) formally modeled the interaction between individuals and a decision maker as a Stackelberg Game, and proposed a framework for strategic classification. While taking the individuals' best response into account, the decision maker can make the optimal decision by anticipating strategic manipulation. During recent years, more complex models on strategic classification have been proposed (Ben-Porat & Tennenholtz, 2017; Dong et al., 2018; Braverman & Garg, 2020; Jagadeesan et al., 2021; Izzo et al., 2021; Ahmadi et al., 2021; Tang et al., 2021; Zhang et al., 2020b; 2022; Eilat et al., 2022; Liu et al., 2022; Lechner & Urner, 2022; Chen et al., 2020b; Xie & Zhang, 2024a). Ben-Porat & Tennenholtz (2017) developed a best response linear regression predictor where two players compete and each gets a payoff depending on the proportion of the points he/she predicts more accurately than the other player. Dong et al. (2018) focused on the online version of the strategic classification algorithm. Chen et al. (2020b) developed a strategic-aware linear classifier to minimize the Stacelberg regret. Tang et al. (2021) considered the setting where the decision maker only knew a subset of individuals' actions. Levanon & Rosenfeld (2022) generalized strategic classification to situations where individuals and the decision maker have aligned interests. Lechner & Urner (2022) proposed a novel loss function considering both the accuracy of the prediction rule and its vulnerability to strategic manipulation. Eilat et al. (2022) relaxed the assumption that individual best responses are independent of each other and proposed a robust learning framework based on a Graph Neural Network. Horowitz et al. (2024) considered the self-selection problem in strategic classification where agents decide whether to participate in the system. Xie & Zhang (2024b) proposed a welfare-aware optimization framework in strategic learning settings.

Regarding randomness, Braverman & Garg (2020) proposed that randomized linear classifiers can be more robust to strategic behaviors. Moreover, Jagadeesan et al. (2021) added noise to standard strategic classification and modified the *standard microfoundations* into *alternative microfoundations* to let a portion of individuals be irrational and not have perfect knowledge about the decision maker's policy. However, they just consider directly adding noise to the best response without modeling the unforeseeable and imitative nature of human agents' behavior.

### A.2 Improvement with a label change

Another line of research takes improvement into account(Liu et al., 2018; Zhang et al., 2020b; Liu et al., 2020; Rosenfeld et al., 2020; Chen et al., 2020a; Haghtalab et al., 2020; Kleinberg & Raghavan, 2020; Alon et al., 2020; Miller et al., 2020; Shavit et al., 2020; Bechavod et al., 2021; Jin et al., 2022; Barsotti et al., 2022; Ahmadi et al., 2022a; Raab & Liu, 2021; Heidari et al., 2019; Somerstep et al., 2024). Liu et al. (2018; 2020); Zhang et al. (2020b); Rosenfeld et al. (2020); Ahmadi et al. (2022b) studied the conditions under which individuals will choose to improve their qualifications. Specifically, Liu et al. (2018) investigated how different decision rules (e.g. maxutil, fair) influence population qualification. Liu et al. (2020) modeled the improvement cost as a random variable and further pointed out that a subsidizing mechanism for individual costs can be beneficial for improving behaviors. Zhang et al. (2020b) studied the dynamic of population qualification under a partially observed Markov decision problem setting, where improvement probability is given as a parameter. Rosenfeld et al. (2020) proposed a *Look-ahead regularization* to directly penalize the drop of population qualification. Ahmadi et al. (2022b) proposed a *common improvement capacity model* and a *individualized improvement capacity model* to optimize social welfare and fairness while considering individual improvement. Jin et al. (2022); Chen et al. (2023) focused on designing subsidy mechanisms to incentivize improvement.

### A.3 Studies considering both behaviors

There are other studies considering both strategic manipulation and improvement, but most of them modeled manipulation and improvement in a similar way(Chen et al., 2020a; Haghtalab et al., 2020; Kleinberg & Raghavan, 2020; Alon et al., 2020; Miller et al., 2020; Shavit et al., 2020; Bechavod et al., 2021; Jin et al., 2022; Barsotti et al., 2022; Ahmadi et al., 2022a; Harris et al., 2022b; Horowitz & Rosenfeld, 2023; Yan et al., 2023; Chen et al., 2023). Specifically, as illustrated in the abstract, the above-listed works all assume

the agents can foresee the outcomes of their actions and they can change their features arbitrarily as long as the cost function permits. Moreover, these works modeled improvement and manipulation by assigning causal/non-causal features, which may not cover the practical cases in our paper such as college admission and job application. We further compare the differences between our model and causal strategic learning in App. B.1. Additionally, Perdomo et al. (2020); Izzo et al. (2021); Hardt et al. (2022); Jin et al. (2024) proposed and elaborated the concept of *performative prediction* where predictive decisions can influence the outcomes to predict. In general, this framework can model both manipulation and improvement but lacks interpretability.

There are only a few works considering both actions while incorporating randomness (Harris et al., 2022a; Bracale et al., 2024). Harris et al. (2022a) mainly focused on the situation where the decision-maker can choose from disclosing partial information of the model (namely, persuasion) and the agents estimate the best response using the partial information. Since the randomness completely comes from the information revealed by the decision-maker, this setting is different from ours where the randomness comes from the unforeseeable and imitative nature of agent improvement and manipulation. As a concurrent work, Bracale et al. (2024) considered a general setting where different actions incur random costs and lead to different feature-label distributions. However, they only focused on estimating the distribution map generally, which is tangential to our focus on disincentivizing manipulation, incentivizing improvement, and promoting fairness. Their contribution to estimating the distribution map may further facilitate the practical applications of our model.

### A.4   Machine learning fairness

While machine learning algorithms are able to achieve high accuracy in different tasks, they are likely to be unfair to individuals from different ethnic groups. To measure the fairness of algorithms, various metrics have been proposed including *demographic parity* (Feldman et al., 2015), *equal opportunity* (Hardt et al., 2016b), *equalized odds* (Hardt et al., 2016b) and *equal resource* (Gupta et al., 2019).

More importantly, several works have studied how strategic behaviors impact fairness (Liu et al., 2018; Zhang et al., 2020b; Liu et al., 2020; Zhang et al., 2022). Specifically, Liu et al. (2018) considered one-step feedback where static fairness does not promote dynamic fairness. Zhang et al. (2020b) analyzed the long-term impact of static fairness metrics based on dynamics of population qualification. Liu et al. (2020) studied how heterogeneity across groups and the lack of realizability can destroy long-term fairness in strategic classification. Zhang et al. (2022) has proposed a probabilistic model to demonstrate strategic manipulation as well as the fairness impacts of strategic behaviors, where the individuals shift their feature distribution instead of directly changing their features. The work also assumed randomness in manipulation cost. Meanwhile, it explored influences on different fairness metrics when strategic manipulation is present(Barocas et al., 2019; Hardt et al., 2016b).

## B   Additional discussions

### B.1   The comparison between our model and causal strategic learning

Previous works in *causal strategic learning* model every strategic classification problem as a *structural causal model* (SCM). SCM is a graphic model depicting the causal relationships between different features and the label, where features can be classified as causal or non-causal after a causal discovery process (Miller et al., 2020). Strategic manipulation means intervening in the non-causal nodes and improvement corresponds to intervening in the causal nodes. Though the model takes both behaviors into account and can accommodate complex causal structures, it has the following weaknesses: (i) The individuals can intervene in any feature node arbitrarily with a deterministic outcome to any value once their budgets permit, which is not practical as illustrated in 1; (ii) In most real-world cases, individuals are not able to intervene the observable features directly. Instead, they intervene in other unobserved features (causal or non-causal) to change the observable features. So it is sometimes meaningless to distinguish whether an observable feature is causal or non-causal, because the root causes of its value change may be diverse.

We illustrate (ii) more clearly in Fig. 7, a causal graph where $U, V$ are unobserved. However, $U$ is non-causal and $V$ is causal. It is easy to see only $X$ is observable and correlated to $Y$, but its change can be either "causal" or "non-causal" with respect to $Y$.

By contrast, our probabilistic framework does not classify $X$ as causal or non-causal. It models both manipulation and improvement as imitating qualified individuals and incorporates the randomness of outcomes and costs. With limited control over their features, individuals can only expect a distribution shift and may even fail when they take certain actions. We believe the concise yet effective design of our model is more suitable for many practical situations nowadays, while the causal strategic models sometimes assign too much power to individuals.

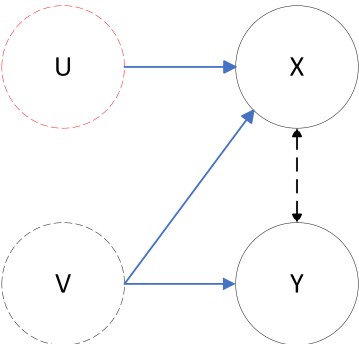

Figure 7: An example causal graph where only $X$ is observable and $U, V$ are unobserved.

## B.2 More practical examples fitting to our model

In Sec. 1 and Appendix B.1, we already explain the motivation of our model in detail. Here we provide more motivating examples besides *college admission*:

1. *Loan application*:

   (a) Manipulation: an unqualified applicant may "steal" the features from qualified ones by purchasing a social security card (SSN) from the hackers. The "stolen" features are still random when the applicant decides to purchase an SSN because the card is often randomly drawn from many stolen cards of qualified individuals.

   (b) Improvement: an unqualified applicant may observe the qualified individuals' profiles and strive to imitate their behaviors. However, the applicant never knows the realization of his/her features before trying to improve. The applicant can only try their best to mimic qualified individuals and expects the successful imitation will cause his/her feature distribution to shift.

2. *Job application*:

   (a) Manipulation: an unqualified applicant may "steal" the features from qualified ones by hiring an imposter to take the interview instead of him/her (especially when remote interviews are prevalent today). Similar to previous examples, the applicant does not know the exact feature realization when making the decision to manipulate.

   (b) Improvement: an unqualified applicant may still observe the features of qualified ones by reading their interview preparation tips or looking at their technical portfolios. Then they may try hard to imitate the qualified individuals. Similar to previous examples, the applicant still has no idea of the exact outcome when he/she decides to improve.

## B.3 The option of "doing nothing"

Our model implicitly includes the action of "doing nothing". Specifically, our model assumes improvement only succeeds with probability $q$; and the improvement cost differs among agents which is modeled as a random variable $C_I$. For agents who do nothing, we may consider them as those who take improvement action at **zero costs** but **fail to improve**. Because no cost is incurred and the improvement fails, the resulting features remain the same and the outcome is equivalent to "doing nothing".

Although it may be intuitive to model "doing nothing" as a separate action, we argue that it is more realistic in practice for individuals to always take an action. As pointed out by Horowitz et al. (2024), agents who decide

to participate in a system will pay an unavoidable cost, and the cost is a part of the "improvement cost". On the contrary and as we argued before, unqualified individuals in our model are not given an explicit option of "doing nothing" with a deterministic 0 cost and 0 utility in the first place. In our framework where improvement is modeled as the imitative behavior in social learning (the third paragraph on page 2), all unqualified individuals staying in the system are influenced by the desire to be qualified and must do something to imitate the qualified profiles because they know they are unqualified. Otherwise, they will just quit the system and are not in our interest. In the college admission example, all unqualified students who do not give up applications and do not cheat are doing similar things: striving to learn more courses, take tests, and prepare for application packages. The only difference is the realization of the improvement outcomes (some students really improve, some do not). To concretely answer your question of $C_M, C_I$, we slightly modify the "budget" to "utility" in footnote 3: For all agents already in the decision-making system, they already have a sufficiently large initial utility exceeding $C_M, C_I$. This means if they choose to quit, they lose the initial utility.

### B.4 Discussion on adjusted preferences

**Utility loss from the adjusted preferences.** Although adjusting preferences is a simple yet effective way to promote fairness and disincentivize manipulation, the actual utility received by the decision-maker inevitably diminishes as $k_1$ or $k_2$ changes (as the actual utility the decision-maker receives is always determined by the original function $U(\theta)$ in equation 3). Nonetheless, such diminished utilities may still be higher than the utility under non-strategic policy $\widehat{\theta}^*$. This is illustrated empirically in Sec. 5 and Appendix C.

**Adjusted preference as a regularizer to promote fairness.** We have shown that adjusting weights $k_1, k_2, k_3$ in learning objective (equation 5) can control the individual behavior and algorithmic fairness. Indeed, we can view this adjustment mechanism as a *regularization* method: by adjusting weights, we are essentially changing the objective $U(\theta)$ by adding a regularizer, i.e.,

$$\widehat{U}(\theta) + \Phi(\theta, k_1, k_2, k_3) = U(\theta) + \underbrace{\Delta\Phi(\theta, k_1, k_2, k_3)}_{\textbf{regularizer}}$$

with the regularizer $\Delta\Phi(\theta, k_1, k_2, k_3)$ defined as follows:

$$\Phi(\theta, k_1, k_2, k_3) - \Phi(\theta, u(1-\alpha), u(1-\alpha), u(1-\alpha))$$

Weights $k_1, k_2, k_3$ are the regularization parameters. The analysis in Sec. 4.2 and 4.3 suggests that to learn optimal policies that satisfy certain constraints such as bounded fairness violation and/or bounded individual's manipulation, we may transform this *constrained* optimization into a *regularized unconstrained* optimization. This view, by incorporating fairness and strategic classification in a simple unified framework, may provide insights for researchers from both communities.

### B.5 Estimate Model Parameters

**A complete estimation procedure.** With only the knowledge of conditional distribution of qualified individuals $P_{X|Y}(x|1)$ and the population's qualification rate $\alpha$, we introduce a complete procedure to estimate $P_{X|Y}(x|0), q, P^I, \epsilon, P_{C_M - C_I}(x)$ sequentially. Specifically, we need to do controlled intervention experiments on an experimental population as follows.

1. Estimate $P_{X|Y}(x|0)$: Set the lowest decision threshold $\theta = 0$ to estimate $P_{X|Y}(x|0)$. Since all unqualified individuals will be accepted, the resulting distribution is the original mixture distribution $(1-\alpha) \cdot P_{X|Y}(x|0) + \alpha \cdot P_{X|Y}(x|1)$. Thus, with minor assumptions on the feature distribution families, we can estimate $P_{X|Y}(x|0)$.

2. Estimate $q$: Apply the strictest auditing procedures (e.g., audit everyone in [26]) to the population to disable manipulation. With manipulation disabled and arbitrary decision threshold $\theta$ applied, all unqualified people choose to improve, and the resulting qualification rate is $(1-\alpha)q + \alpha$. Thus, by examining the qualification rate after the intervention we can get the estimation of $q$.

3. Estimate $P^I$: Apply an arbitrary decision threshold $\theta$ to the population, the resulting population probability density distribution will be a mixture of $(1-\alpha)(1-q)P^I + [(1-\alpha)q + \alpha]P_{X|1}$. Similarly, with minor assumptions on the distribution family of $P^I$, we can estimate $P^I$.

4. Estimate $\epsilon$: With $q, P^I$ known, the decision-maker can first apply another arbitrary $\theta$ to new samples from the population and observe the resulting new population. This gives the new qualification rate $\alpha_p$. Because $\alpha_p = \alpha + (1-\alpha)(1 - P_M(\theta)q)$ where $P_M(\theta)$ is the probability of manipulation under $\theta$, we can then compute the value of $P_M(\theta)$. Note that the decision-maker also knows how many individuals (among all individuals) are discovered to manipulate (cheat), and let this proportion be $\epsilon_c$, then we can estimate the manipulation detection probability $\epsilon$ as $\frac{\epsilon_c}{P_M(\theta)}$.

5. Finally, with all previous parameters known, we can apply different $\theta$ to the population several times to obtain data points of $P_M$. Then since $P_M$ corresponds to points of $F_{C_M - C_I}$, with minor assumptions on the distribution family of $P_M$, we can directly fit the distribution and get $P_{C_M - C_I}$.

It is worth noting that all the above steps can be more robust by doing multiple intervention experiments, and controlled experiments are necessary Miller et al. (2020). We will add the above discussion to the paper to improve its significance. Finally, with all the parameters, the decision-maker can first apply Thm. 4.6 to see how to adjust its preferences, and then perform a grid search to find the best $k$.

**Robustness of results when $q, \epsilon$ are noisy.** We also present an experiment to relax the assumption that the decision-maker knows $q, \epsilon$ exactly on FICO data. Instead, they only know $q + \delta$ or $\epsilon + \delta$ where $\delta$ is a Gaussian noise. Table 3 show the results with noisy $q$, while Table 4 show the results with noisy $\epsilon$). The results show adjusting $k$ still works under noisy $q$ and $\epsilon$ although inconsistency exists.

Table 3: Comparison between three types of optimal thresholds (FICO data) when there is a Gaussian noise on $q$ with standard deviation 0.1 and $k_{1,c} = k_{1,aa} = 1.25$. For utility and $P_M$, the left value in parenthesis is for Group $a$, while the right is for Group $b$. The fairness metric is *eqopt*.

| Threshold category | Utility | $P_M$ | Unfairness |
|---|---|---|---|
| Non-strategic | $(0.698, 0.171)$ | $(0.331, 0.513)$ | 0.136 |
| Original average **noisy** strategic | $(0.703, 0.201)$ | $(0.212, 0.284)$ | 0.057 |
| Adjusted average **noisy** strategic | $(0.700, 0.192)$ | $(0.170, 0.220)$ | 0.043 |

Table 4: Comparison between three types of optimal thresholds (FICO data) when there is a Gaussian noise on $\epsilon$ with standard deviation 0.1 and other settings stay the same.

| Threshold category | Utility | $P_M$ | Unfairness |
|---|---|---|---|
| Non-strategic | $(0.698, 0.171)$ | $(0.331, 0.513)$ | 0.136 |
| Original average **noisy** strategic | $(0.700, 0.195)$ | $(0.192, 0.251)$ | 0.050 |
| Adjusted average **noisy** strategic | $(0.698, 0.185)$ | $(0.158, 0.194)$ | 0.037 |

## C  Additional empirical results

### C.1  Additional results on FICO score

Firstly, Fig. 9 shows the conditional distribution $P_{X|YS}$ and $P^I$ of each ethnic group. Fig. 8 demonstrates Assumption 2.1 is satisfied. Fig. 21 shows the (non)-strategic optimal thresholds under different combinations of $q, \epsilon$ for each ethnic group. All four plots demonstrate the correctness of Thm. 4.1.

**Experiments with demographic parity as a new fairness metric.**

We also reconducted the above experiments with demographic parity (DP) as the new fairness metric. As illustrated in Sec. 4.3, $P_s^{DP}(x) = P_{X|S}(x|s)$. Similar to Fig. 4 and Fig. 4, we produce Fig. 11 based on DP, which demonstrate the same patterns as the figures based on Eqopt.

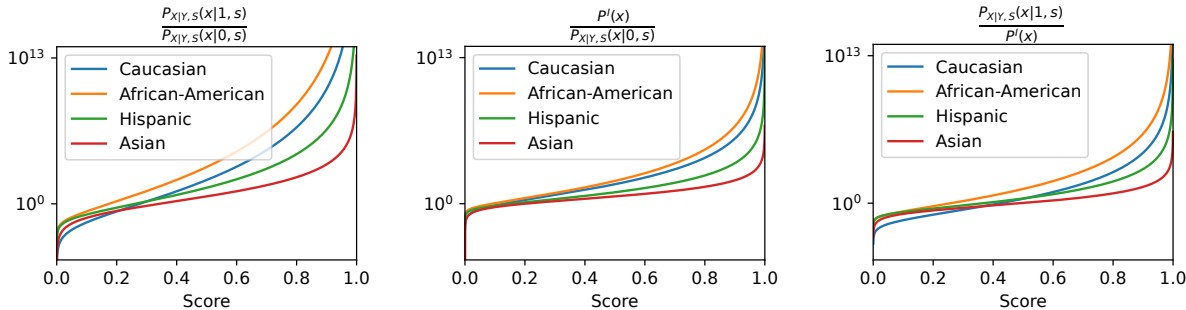

Figure 8: Illustration of Assumption 2.1 on FICO Data

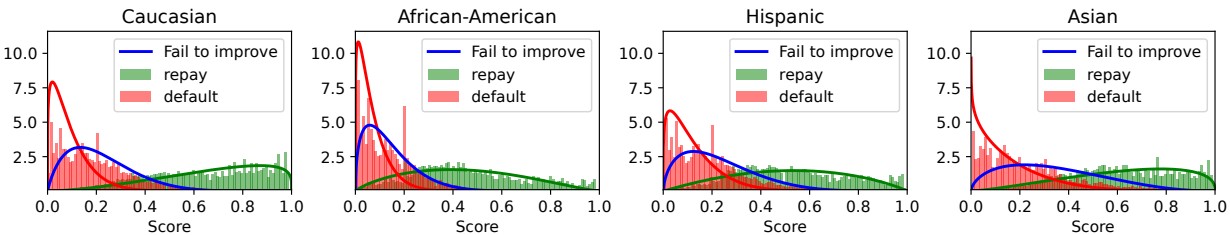

Figure 9: Manipulation curve and manipulation probability for both groups under optimal non-strategic thresholds

## C.2  Results for Gaussian Data

Assume there are two groups For $s \in \{a, b\}$, we both have:

$$
\begin{aligned}
P_{X|YS}(x|0,s) &\sim \mathcal{N}(0,1) \\
P_s^I &\sim \mathcal{N}(0.5,1) \\
P_{X|YS}(x|1,s) &\sim \mathcal{N}(1,1) \\
C_M - C_I &\sim \mathcal{N}(0,0.25)
\end{aligned}
\tag{6}
$$

We first illustrate the conditional feature distributions for Gaussian data in Fig. 10. With these parameters pre-determined, we still need to vary $\alpha, \epsilon, q$ to obtain $\widehat{\theta}^*, \theta^*$ under different parameter combinations.

### (Non)-strategic optimal threshold and utility

To illustrate the complex nature under different permutations of parameters, with the pre-determined parameters in equation 6 and $\alpha = 0.6$, we vary $q$ and plot both non-strategic optimal thresholds and regular strategic ones with respect to different $\epsilon$ as shown in the bottom plot of Fig. 12, where the lower graphs illustrate Thm. 4.1, i.e. the red line is always under the blue line.

We also demonstrate the strategic utility under different combinations of $q, \epsilon$ with pre-determined parameters in equation 6 and $\alpha = 0.3$ or $\alpha = 0.6$. Fig. 13 and 14 suggest the complicated nature of regular strategic utility under different parameter combinations. It is possible to have 0,1 or 2 extreme points.

### Illustration of threshold shifts while adjusting $k$

To illustrate 4.5, we demonstrate the effects of adjusting each of $k_1, k_2, k_3$. According to Fig. 15 and 16, we can see when $k_1$ is large enough, the optimal strategic threshold is definitely lower than the optimal non-strategic ones. However, when $\alpha$ is small, we need larger $k_1$ to pull $\theta^*$ downward. According to Fig. 17 and 18, we can see when the population is majority qualified, adjusting $k_2$ is not guaranteed to shift $\theta^*$ upward (Fig. 18).

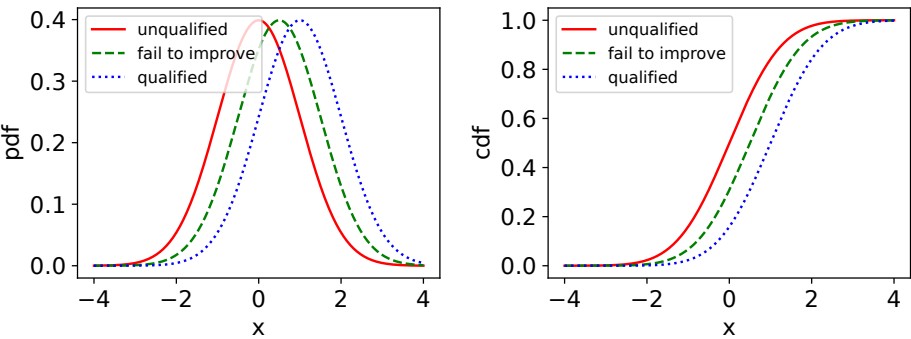

Figure 10: Illustration of equation 6

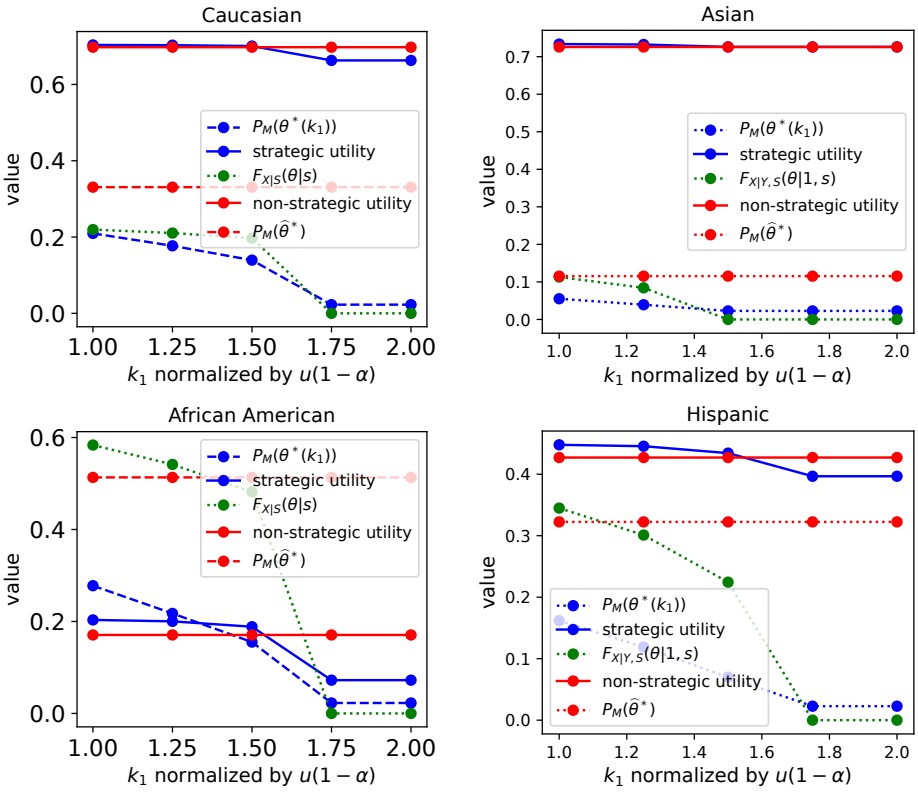

Figure 11: Illustration of Thm. 4.5 and Thm. 4.6 in FICO Data with fairness metric DP. Left figure is for Caucasian and African American, while the right is for Asian and Hispanic

**Illustration of condition 1.(i), Thm. 4.5**

We first show a parameter setting satisfying condition *1.(i)* in Thm. 4.5. With pre-determined parameters in equation 6 , we set $q = \epsilon = 0.5$ and $\alpha_a = 0.2, \alpha_b = 0.25$. This matches the notation tradition in Sec. 4.3 where group $a$ is the disadvantaged group with a lower qualified percentage. Also, because $q + \epsilon \geq 1$, the setting satisfies condition *1.(i)* in Thm. 4.5. We first illustrate the manipulation probability under optimal original strategic threshold $\theta_s^*$ as in Fig. 10. From Fig. 20, we can set $k_{1a} = k_{2a} = 1.25$ to let the strategic utility still be larger than the one under non-strategic optimal threshold(i.e. the solid blue line is above the solid red line), while lower the cumulative density dramatically (i.e. the dotted green line) to admit more

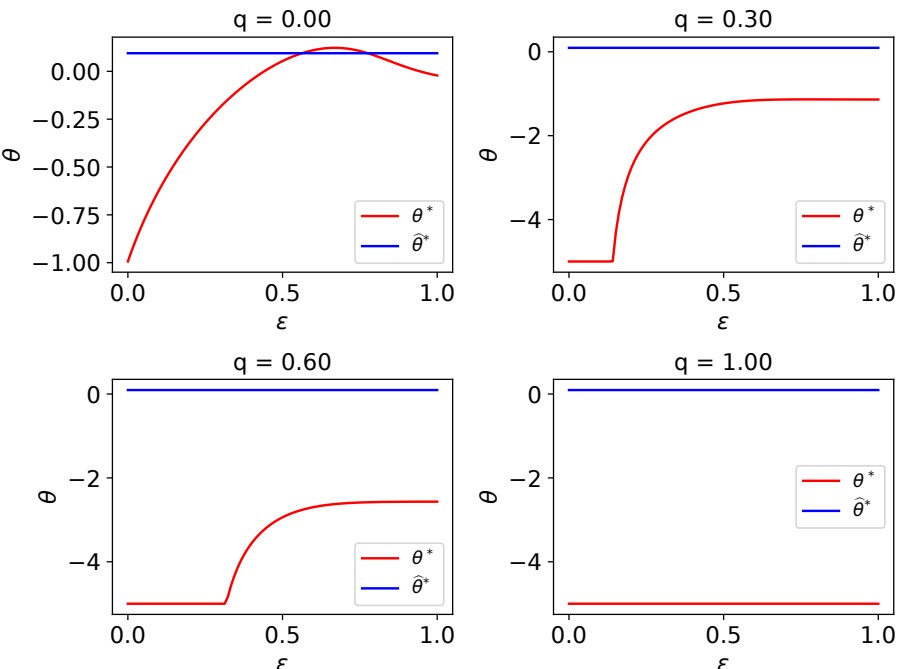

Figure 12: (Non)-strategic optimal threshold. We regularize $y$-axis to be [-5,5] to prevent numerical issues.

qualified individuals and disincentivize manipulation (i.e. the dashed blue line). The details of comparisons are shown in Table 6.

**Illustration of condition 1.(ii), Thm. 4.5**

With pre-determined parameters in equation 6 , we set $q = \epsilon = 0.25$ and $\alpha_a = 0.4, \alpha_b = 0.6$. This matches the notation tradition in Sec. 4.3 where group $a$ is the disadvantaged group with a lower qualified percentage. We first illustrate the manipulation probability under optimal original strategic threshold $\theta_s^*$ as in Fig. 19. Fig. 19 reveals that *1.(ii)* in 4.5 is satisfied because the orange and green points are both located before the extreme large point of $P_M(\theta)$. Thus, we could increase $k_{1s}$ to disincentivize manipulation while improving fairness as shown in Fig. 20. From Fig. 20, we can set $k_{1a} = k_{2a} = 1.25$ to let the strategic utility still be larger than the one under non-strategic optimal threshold(i.e. the solid blue line is above the solid red line), while lower the cumulative density dramatically (i.e. the dotted green line) to admit more qualified individuals and disincentivize manipulation (i.e. the dashed blue line). In Table 5, We summarize the comparison between non-strategic $\widehat{\theta}^*$, original strategic $\theta^*$, and adjusted strategic $\theta^*(k_1)$ (when $k_{1,c} = k_{1,aa} = 1.25$). It shows that decision-makers by adjusting preferences can significantly mitigate unfairness and disincentivize manipulation, with only slight decreases in utilities.

**Illustration of condition 2, Thm. 4.5**

Besides, we also show one more parameter setting satisfying condition 2 in Thm. 4.5. With pre-determined parameters in equation 6 , we also set $q = \epsilon = 0.2$ and $\alpha_a = 0.3, \alpha_b = 0.35$. This matches the notation tradition in Sec. 4.3 where group $a$ is the disadvantaged group with a lower qualified percentage. Also, based on Fig. 19, $q + \epsilon < 1$ and $\alpha_a, \alpha_b < 0.5$, the setting satisfies condition 2 in Thm. 4.5. We first illustrates the manipulation probability under optimal original strategic threshold $\theta_s^*$ and non-strategic threshold $\widehat{\theta}_s^*$ as in Fig. 19. As shown in Fig. 20, for both groups, we demonstrate the manipulation probability for $\widehat{\theta}^*, \theta^*$ and $\theta(k_1)$ when $k_1$ varies, (non)-strategic utility and cumulative density conditioned on $Y = 1$ (i.e. this measures the unfairness based on *equal opportunity*). This plot suggests we can find suitable $k_{2a}$ and $k_{2b}$ to disincentivize manipulation and promote fairness, while also making the utility higher than the one under non-strategic optimal threshold. From Fig. 20, we can set both $k_{2a}$ and $k_{2b}$ at 1.25 to let the strategic utility

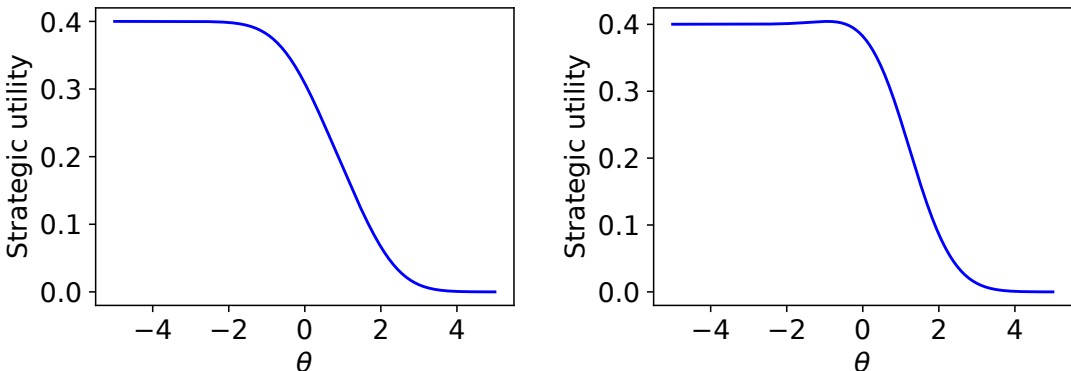

Figure 13: Regular strategic utility when $\alpha = 0.6$. The left figure has $\epsilon = 0, q = 0.5$ and the right has $\epsilon = 0.75, q = 0.25$

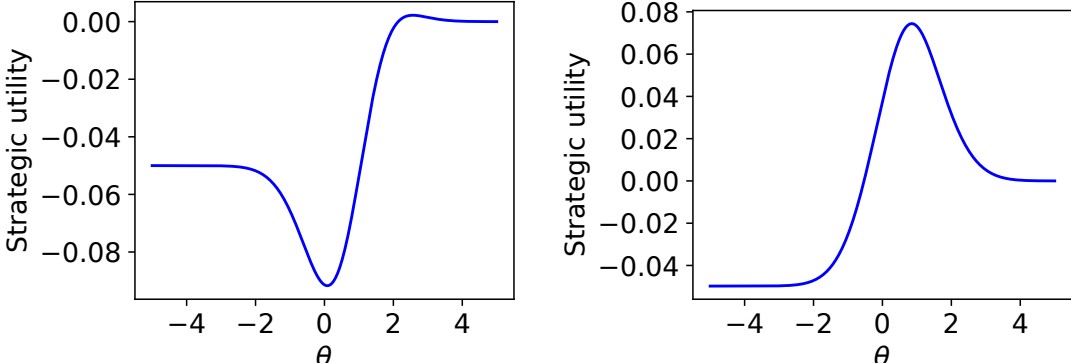

Figure 14: Regular strategic utility when $\alpha = 0.3$. The left figure has $\epsilon = 0, q = 0.5$ and the right has $\epsilon = 0.75, q = 0.25$

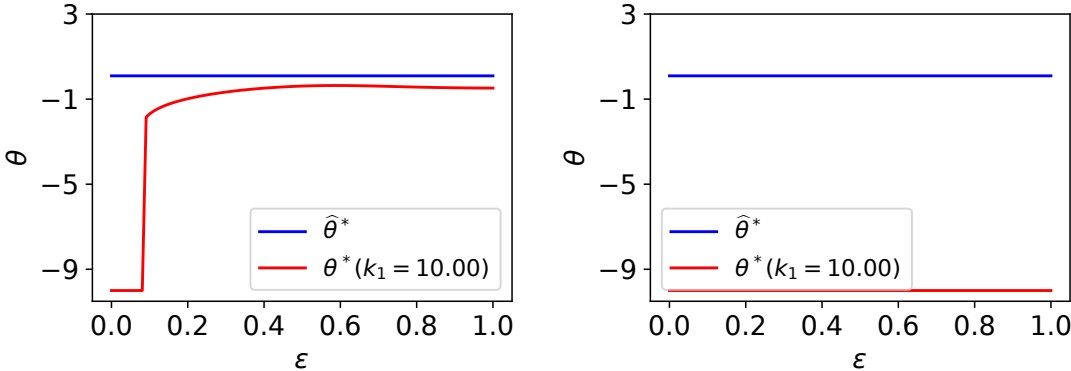

Figure 15: Strategic optimal threshold $\theta^*(k_1)$ after increasing $k_1$ while keeping $k_2, k_3$ fixed. Left figure has $q = 0.01$ and right figure has $q = 0.99$, while both figures have $\alpha = 0.6$

still be larger than the utility under non-strategic optimal threshold (i.e. the solid blue line is above the solid red line), while keeping the cumulative density function closer (i.e. the green dotted line) to mitigate unfairness, and also disincentivize manipulation (i.e. the blue dashed line). The details of comparisons are shown in Table 7.

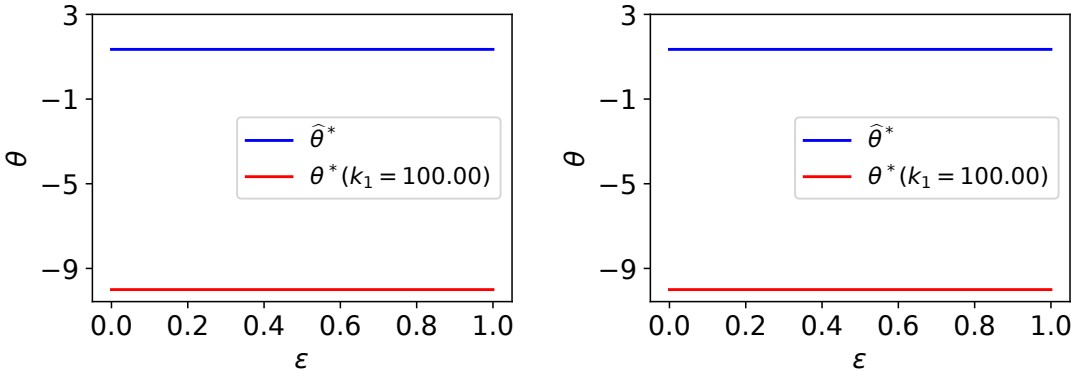

Figure 16: Strategic optimal threshold $\theta^*(k_1)$ after increasing $k_1$ while keeping $k_2, k_3$ fixed. Left figure has $q = 0.01$ and right figure has $q = 0.99$, while both figures have $\alpha = 0.3$. We regularize $y$-axis to be [-5,5] to prevent numerical issues.

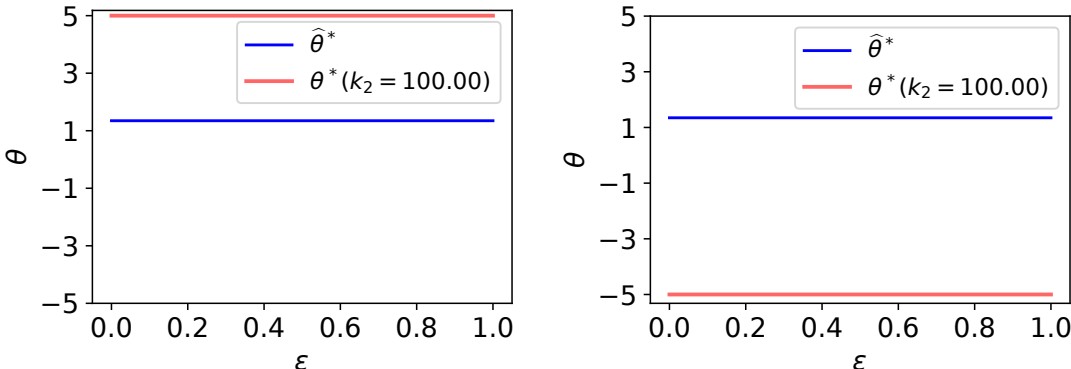

Figure 17: Strategic optimal threshold $\theta^*(k_2)$ after increasing $k_2$ while keeping $k_1, k_3$ fixed. Left figure has $q = 0.01$ and right figure has $q = 0.99$, while both figures have $\alpha = 0.3$. We regularize $y$-axis to be [-5,5] to prevent numerical issues.

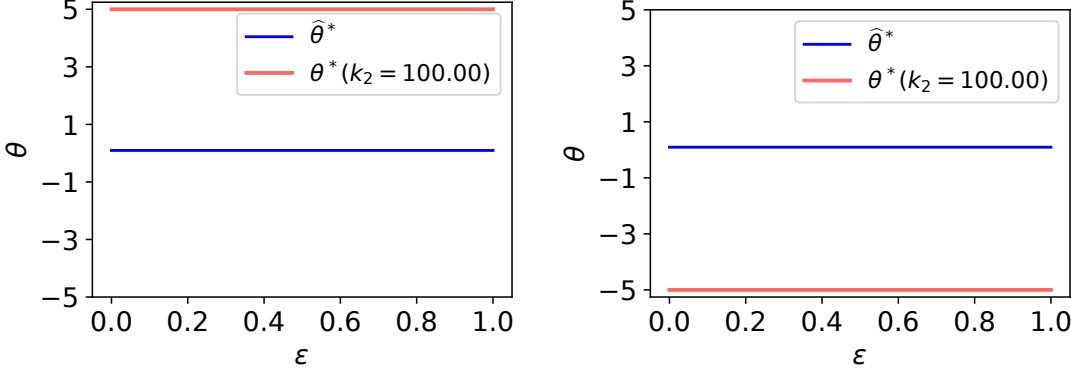

Figure 18: Strategic optimal threshold $\theta^*(k_2)$ after increasing $k_2$ while keeping $k_1, k_3$ fixed. Left figure has $q = 0.01$ and right figure has $q = 0.99$, while both figures have $\alpha = 0.6$

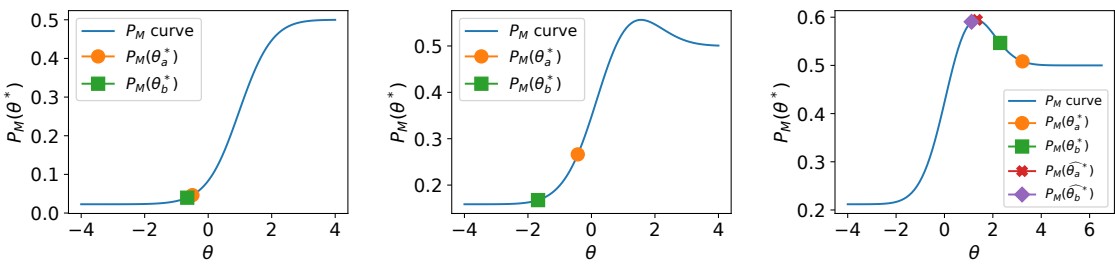

Figure 19: Manipulation probability $P_M(\theta)$: from left to right are plots for condition *1.(i), 1.(ii), 2*

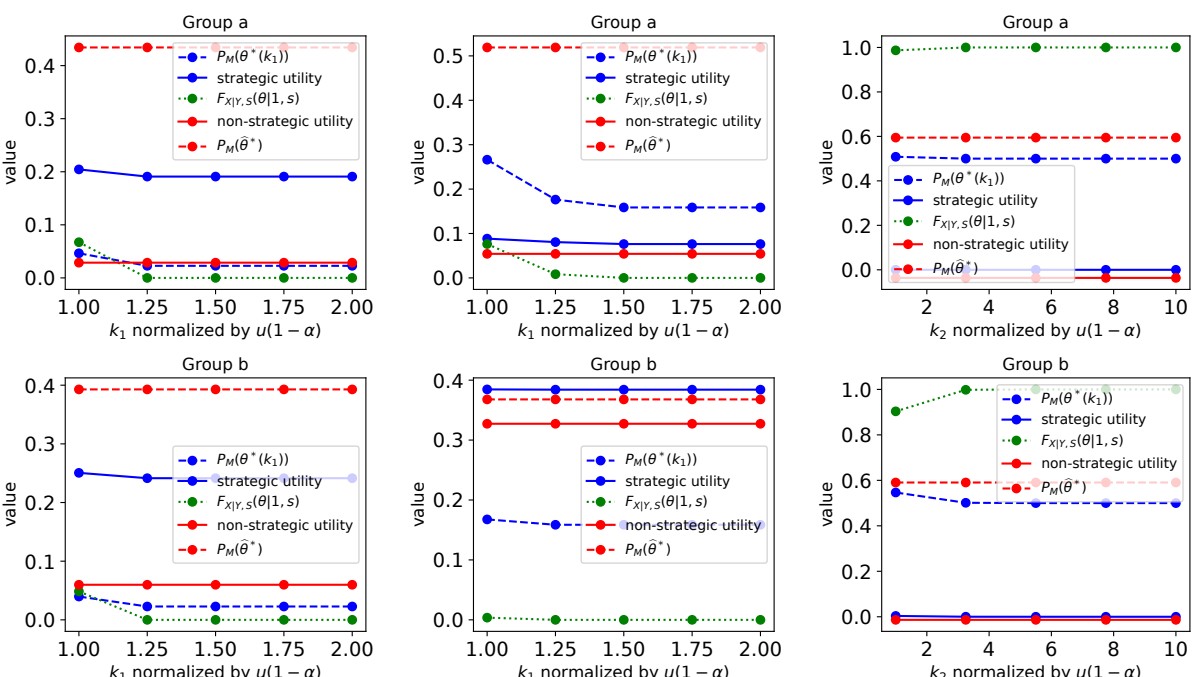

Figure 20: Illustration of Thm. 4.5 and Thm. 4.6. From left to right are illustrations for condition *1.(i),1.(ii),2*

Table 5: Comparison between three types of optimal thresholds for Gaussian data satisfying condition *1.(i)*. For utility and $P_M$, the left value in parenthesis is for Group $a$, while the right is for Group $b$. The fairness metric is *eqopt*.

| Threshold category | Utility | $P_M$ | Unfairness |
|---|---|---|---|
| Non-strategic | $(0.054, 0.327)$ | $(0.519, 0.368)$ | $0.280$ |
| Original strategic | $(0.081, 0.384)$ | $(0.266, 0.168)$ | $0.073$ |
| Adjusted strategic | $(0.088, 0.385)$ | $(0.176, 0.159)$ | $0.008$ |

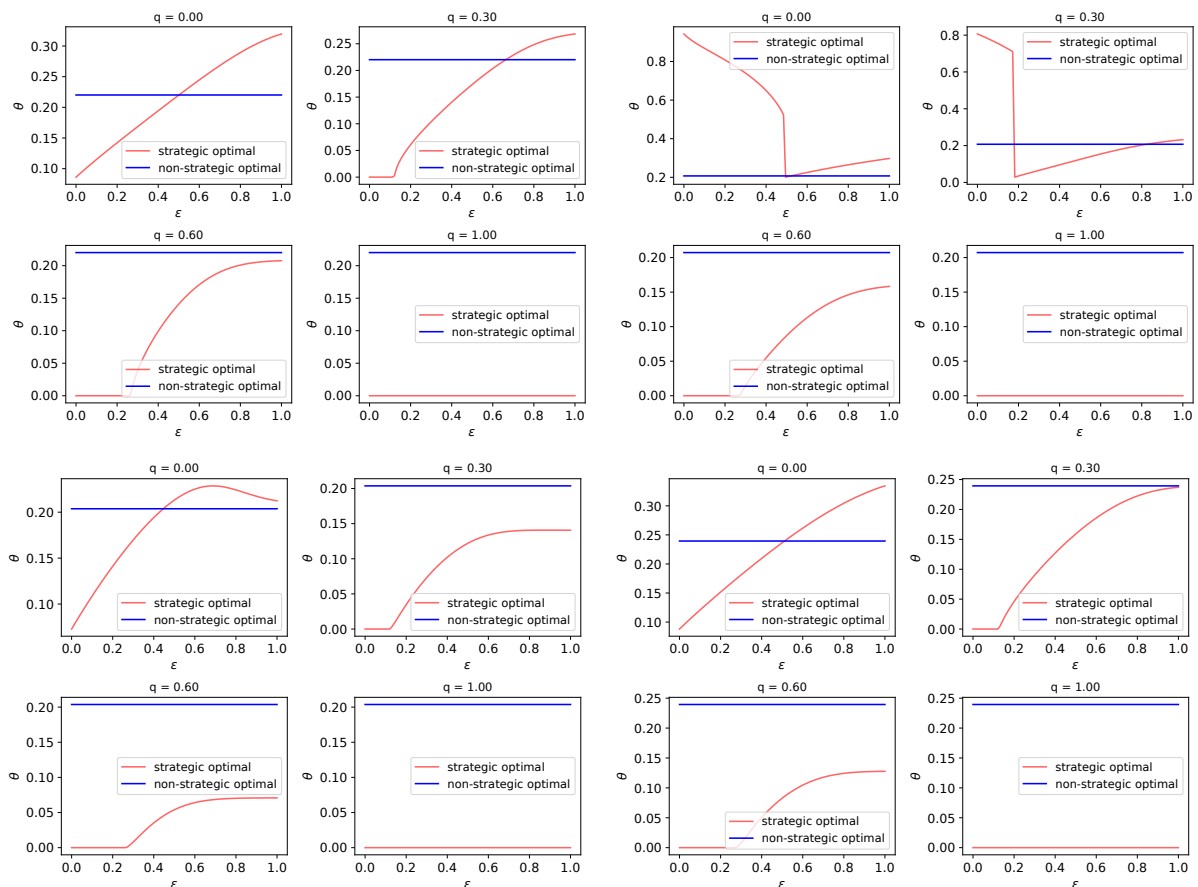

Figure 21: (Non)-strategic optimal thresholds under different $q, \epsilon$ for different ethnic groups (top left: Caucasian; top right: African American; bottom left: Asian; bottom right: Hispanic)

Table 6: Comparison between three types of optimal thresholds for Gaussian data satisfying condition *1.(ii)*. For utility and $P_M$, the left value in parenthesis is for Group $a$, while the right is for Group $b$. The fairness metric is *eqopt*.

| Threshold category | Utility | $P_M$ | Unfairness |
|---|---|---|---|
| Non-strategic | $(0.029, 0.060)$ | $(0.434, 0.393)$ | 0.086 |
| Original strategic | $(0.204, 0.251)$ | $(0.046, 0.040)$ | 0.019 |
| Adjusted strategic | $(0.191, 0.241)$ | $(0.023, 0.023)$ | 0 |

Table 7: Comparison between three types of optimal thresholds for Gaussian data satisfying condition *2*. For utility and $P_M$, the left value in parenthesis is for Group $a$, while the right is for Group $b$. The fairness metric is *eqopt*.

| Threshold category | Utility | $P_M$ | Unfairness |
|---|---|---|---|
| Non-strategic | $(-0.036, -0.014)$ | $(0.674, 0.686)$ | 0.088 |
| Original strategic | $(0.001, 0.004)$ | $(0.508, 0.547)$ | 0.084 |
| Adjusted strategic | $(0, 0)$ | $(0.500, 0.500)$ | 0.002 |

# D  Derivations and Proofs

## D.1  Derivations of equation 1

$U_M(\theta)$ is the expected utility gain of an unqualified agent if choosing to manipulate: i. If the manipulation is not exposed, the probability of admission is $1 - F_{X|Y}(\theta|1)$ because the manipulation leads the agents to get his/her new feature from $P_{X|Y=1}$, which happens at a probability $1 - \epsilon$; ii. If the manipulation is exposed, the probability of admission is 0, which happens at a probability $\epsilon$; iii. If the agent does not manipulate, the probability of admission is $1 - F_{X|Y}(\theta|0)$ because now his/her feature is from the unqualified population, and keep in mind that the agents will never know the exact values of his/her feature when he/she makes decisions; Then according to the total probability theorem, the expectation of utility gain $U_M(\theta) = (1 - \epsilon) \cdot (1 - F_{X|Y}(\theta|1)) + \epsilon \cdot 0 - (1 - F_{X|Y}(\theta|0)) - C_M$.

$U_I(\theta)$ is the expected utility gain of an unqualified agent if choosing to improve: i. If the improvement succeeds, the probability of admission is $1 - F_{X|Y}(\theta|1)$ because the improvement leads the agents to get his/her new feature from $P_{X|Y=1}$, which happens at a probability $q$; ii. If the manipulation is exposed, the probability of admission is $1 - F^I(\theta)$, which happens at a probability $1 - q$; iii. If the agent does not manipulate, the probability of admission is $1 - F_{X|Y}(\theta|0)$.

Then according to the total probability theorem, we can derive $U_I(\theta)$ as well. Finally, substitute above two terms into $P_M(\theta) = \Pr(U_M(\theta) > U_I(\theta))$ and we get equation 1.

## D.2  Proof of Thm. 2.3

Assumption 2.2 ensures that $P_{C_M - C_I} > 0$ when $\theta$ in its domain. Thus, we can directly take the derivative inside equation 1, we can get $(1 - q) \cdot P^I(\theta) - (1 - q - \epsilon)P_{X|Y}(\theta|1)$. To get its sign, we only need to consider $(1 - q) - (1 - q - \epsilon)\frac{P_{X|Y}(\theta|1)}{P^I(\theta)}$.

Thus, if $1 - q - \epsilon \le 0$, the derivative is always larger than 0 (since $q < 1$). So under this situation, $P_M$ is always increasing. Otherwise, since $\frac{P_{X|Y}(\theta|1)}{P^I(\theta)}$ is increasing according to Assumption 2.1, it will first increase and then decrease, with $\frac{P_{X|Y}(\theta_{max}|1)}{P^I(\theta_{max})} = \frac{1-q}{1-q-\epsilon}$.

Since $\frac{P_{X|Y}(\theta_{max}|1)}{P^I(\theta_{max})}$ is monotonically increasing and $\frac{1-q}{1-q-\epsilon} = 1 + \frac{\epsilon}{1-q-\epsilon}$, when $q$ increases $1 + \frac{\epsilon}{1-q-\epsilon}$ increases, making $\theta_{max}$ increases. The same also holds when $\epsilon$ increases. Note that while $q$ or $\epsilon$ increases, we still need $q + \epsilon \le 1$.

## D.3  Proof of Thm. 4.1

Assume $\theta \in (a, b)$. When $q \to 1$, improvement will always succeed. Also, Thm. 2.3 reveals $P_M(\theta)$ reaches its minimum when $\theta \to a$, so $P_M(a) < 0.5$. Thus, improvement will always bring a benefit that is larger than manipulation to the strategic decision-maker (since improvement always succeeds). Thus, the decision maker may set a threshold as low as possible ($\to a$) to maximize its utility, which will always be lower than the non-strategic optimal threshold.

## D.4  Proof of Prop. 4.2

Assume $\theta \in (a, b)$. Consider the situation when $k_2, k_3$ both stay fixed and $k_1 \to \infty$, $U = \Phi + \hat{U}$ is dominated by $k_1\phi_1$. Noticing $\phi_1$ reaches its maximum when $\theta \to a$, we will also have the new optimal $\theta^*(k_1) \to a$. Since $a$ is the minimum possible value of the threshold, the optimal threshold when $k_a$ is large enough will definitely be smaller than the optimal non-strategic threshold as well as the original optimal strategic threshold.

## D.5  Proof of Prop. 4.3

Assume $\theta \in (a, b)$. Consider the situation when $k_1, k_3$ both stay fixed and $k_2 \to \infty$, $U = \Phi + \hat{U}$ is dominated by $-k_2\phi_2$. $\phi_2 \to 0$ both when $\theta \to b$ or $a$ (i.e. $\phi_2$ reaches its minimum). However, the non-strategic utility should be 0 when $\theta \to b$ but smaller than 0 when $\theta \to a$ if not majority of people are qualified. This will make the new optimal $\theta^*(k_2) \to b$. Since $b$ is the maximum possible value of the threshold, the optimal threshold when $k_2$ is large enough will definitely be larger than the optimal non-strategic threshold as well as the original optimal strategic threshold.

### D.6 Proof of Prop. 4.4

Assume $\theta \in (a, b)$. Consider the situation when $k_1, k_2$ both stay fixed and $k_3 \to b$, $U = \Phi + \hat{U}$ is dominated by $-k_3\phi_3$. Take the derivative of $(1-\epsilon) \cdot (1 - F_{X|Y}(\theta|1)) - (1 - F_{X|Y}(\theta|0))$ (the term multiplied by $P_M$ in $\phi_3$), we get $1 - (1-\epsilon)\frac{P_{X|Y}(X|1)}{P_{X|Y}(X|0)}$. This suggests the term will first increase and then decrease. Thus, the maximizer of $-k_3 \cdot \phi_3 = -k_3 \cdot P_M \cdot (1 - (1-\epsilon)\frac{P_{X|Y}(X|1)}{P_{X|Y}(X|0)})$ will locate before the root of $(1-\epsilon) \cdot (1 - F_{X|Y}(\theta|1)) - (1 - F_{X|Y}(\theta|0))$. Then noticing that increasing $\epsilon$ will lower the value of the root, we can confirm the existence of $\bar{\epsilon}$ to make the root small enough, thereby making the maximizer of $-k_3 \cdot \phi_3$ smaller enough. Then because $U$ is dominated by $-k_3\phi_3$, $\theta^*(k_3)$ will also be small enough.

### D.7 Proof of Thm. 4.5

Assume $\theta \in (a, b)$. Then: 1. Under condition *1.(i)*, Thm. 2.3 shows $P_M(\theta)$ strictly increases. Because increasing $k_1$ will cause $\theta^*(k_1)$ to left shift until approaching $a$, $P_M(\theta^*(k_1))$ will keep decreasing to its minimum value.

2. Under condition *1.(ii)*, Thm. 2.3 shows $P_M(\theta)$ strictly increases before $\theta_{max}$, where $\frac{P_{X|Y}(\theta_{max}|1)}{P^I(\theta_{max})} = \frac{1-q}{1-q-\epsilon}$. Since $\frac{P_{X|Y}(\theta|1)}{P^I(\theta)}$ is increasing, we would know $\theta^* < \theta_{max}$. Because increasing $k_1$ will cause $\theta^*(k_1)$ to left shift until approaching $a$, $P_M(\theta^*(k_1))$ will keep decreasing to its minimum value.

3. Under condition *2*, Thm. 2.3 shows $P_M(\theta)$ strictly decreases after $\theta_{max}$, where $\frac{P_{X|Y}(\theta_{max}|1)}{P^I(\theta_{max})} = \frac{1-q}{1-q-\epsilon}$. Since $\frac{P_{X|Y}(\theta|1)}{P^I(\theta)}$ is increasing, we would know $\theta^* > \theta_{max}$. Because increasing $k_2$ when $\alpha \leq 0.5$ will cause $\theta^*(k_2)$ to right shift until approaching $a$, $P_M(\theta^*(k_2))$ will keep decreasing to $F_{C_M - C_I}(0)$, which is smaller than $P_M(\hat{\theta}^*)$.

### D.8 Proof of Thm. 4.6

Define $\mathbb{F}_s^c$ as some cumulative density function (CDF) associated with fairness metric $C$. The unfairness $\left| \mathbb{E}_{X \sim P_a^C}[\mathbf{1}(x \geq \theta_a)] - \mathbb{E}_{X \sim P_b^C}[\mathbf{1}(x \geq \theta_b)] \right|$ can also be written as $\mathbb{F}_a^c(\theta_a) - \mathbb{F}_a^c(\theta_b)$.

1. Under situation 1, Thm. 4.5 already reveals increasing $k_1$ can disincentivize strategic manipulation. Meanwhile, $\mathbb{F}_s^c(\theta_s^*(k_1))$ will decrease for both groups because $\theta_s^*(k_1)$ decreases for both group. Thus, there must exist $k_{1a}, k_{1b}$ to mitigate the difference between $\mathbb{F}_a^c(\theta_a^*(k_1))$ and $\mathbb{F}_b^c(\theta_b^*(k_1))$, which is promoting the fairness at the same time of disincentivizing manipulation.

2. Under situation 2, Thm. 4.5 already reveals increasing $k_2$ can disincentivize strategic manipulation. Meanwhile, $\mathbb{F}_s^c(\theta_s^*(k_2))$ will increase for both groups because $\theta_s^*(k_2)$ increases for both group. Thus, there must exist $k_{2a}, k_{2b}$ to mitigate the difference between $\mathbb{F}_a^c(\theta_a^*(k_2))$ and $\mathbb{F}_b^c(\theta_b^*(k_2))$, which is promoting the fairness at the same time of disincentivizing manipulation.

3.Under situation 3, Thm. 4.5 already reveals increasing $k_1$ for group $a$ and increasing $k_2$ for group $b$ can disincentivize strategic manipulation. Meanwhile, $\mathbb{F}_s^c(\theta_a^*(k_1))$ will decrease for $a$ and $\mathbb{F}_s^c(\theta_b^*(k_2))$ increase for $b$. Thus, because $a$ is already the disadvantaged group, the difference between $\mathbb{F}_s^c(\theta_a^*(k_1))$ and $\mathbb{F}_s^c(\theta_b^*(k_2))$ will be mitigated, which is promoting the fairness at the same time of disincentivizing manipulation.

### D.9 Proof of Corollary 4.7

Corollary 4.7 can be derived directly from Thm. 4.5 and Thm. 4.6. To recap, Thm. 4.5 identifies all scenarios under which manipulation is guaranteed to be disincentivized via adjusting preferences; Theorem reftheorem:fairness finds all scenarios when promoting fairness and disincentivizing manipulation can be attained simultaneously; Corollary 4.7 emphasizes all scenarios where disincentivizing manipulation does not guarantee fairness improvement.

In Corollary 4.7, to ensure the manipulation to always be disincentivized, both groups $a, b$ should satisfy **either** scenario identified in Thm. 4.5. This results in four possible combinations, and three out of these four are the scenarios found in Thm. 4.6. The left one situation is the case in Corollary 4.7 (group $a$ satisfies condition 2 and group $b$ satisfies condition 1). In this case, group $a$ can be disincentivized only by

increasing $k_2$. However, increasing $k_2$ can only make the decision threshold $\widehat{\theta}_a^*$ higher, which will exacerbate the unfairness (since group $a$ has $\alpha_a < 0.5$, by condition 2.(i)).

