# OpenReview forum: "Learning under Imitative Strategic Behavior with Unforeseeable Outcomes"
_TMLR — Accepted by TMLR_

### Review · Reviewer_69Tp · 2024-08-25

**Summary Of Contributions:**

This paper studies a model of strategic binary classification of agents using a threshold-based classifier, in which unqualified (negative-label) agents can choose to either attempt to *manipulate* their features (succeeding and imitating a qualified candidate), or to *improve* their features (thereby actually becoming a qualified candidate). Each action has a probability of failure: failing to manipulate results in immediate rejection, and failure to improve results in the agent's label not changing (but possibly the features changing, according to a known distribution). The paper then presents various results about how the decision-maker's choice of threshold is related to various other quantities.

**Audience:**

Yes

**Claims And Evidence:**

Yes

**Requested Changes:**

The most crucial change (critical to securing my recommendation) is that I would want to see more discussion of the various assumptions made by the paper and either (1) why they are justified or (2) some analysis of what happens in the case where they are broken (or both).

Relatively minor notes, questions, and nitpicks, in no particular order, that I would like clarification on:
1. Some of the plots (Figs 14, 19, and 20) exhibit very strange "jumpy" behavior in $\theta$ as a function of the given parameter. It seems reasonably likely to me that this behavior is nothing but a function of "numerical precision"-type issues when dealing with thresholds that are 6+ standard deviations away from the means. Perhaps it would be more instructive/illuminating to, for example, set the bounds of the $\theta$ axis at a reasonable range like $[-5, +5]$, to remove these numerical issues.
2. Regarding the assumptions of continuity of various distributions (2.2 and parts of 2.1): are these assumptions made for clarity/simplicity of exposition, or are they somehow technically vital to the results? Phrased another way, could you obtain the same or similar results without these continuity assumptions?
3. In Section 3, what is the use of specifying a number $u$ for the decision-maker's utility of making a correct decision? Isn't it WLOG to assume $u=1$?

**Strengths And Weaknesses:**

I think the paper is well-written, the problem setup is clear, and the results are interesting. I did not fully check the proofs, but the theorem statements and experimental results both seem reasonable to me.

As someone who does not work in the space of strategic classification, I am not well-placed to judge the novelty of the work; I will leave that to the other reviewer(s) who might have more problem-specific knowledge.

My main criticism is that some of the assumptions of the paper seem to be not sufficiently justified:

1. In Footnote 3/Appendix B.3 it states that agents will always choose to either manipulate or improve. But this does not seem true to me, even in the model studied in the paper: for example, if $C_M, C_I > 1$ then clearly the agent should choose to do neither. If the assumption is that $C_M$ and $C_I$ are always small enough that at least one of the two actions is preferable to doing nothing, this should be stated (and perhaps analyzed---for example, in what regime of $C_M, C_I$ is this the case?)
2. Why aren't agents allowed to know their own feature $x$? In real situations, it seems logical for e.g. students to be able to know roughly how well they will do on an exam, e.g., by taking practice exams. Similarly, it is strange to me that the agents' ability to improve is completely decoupled from their current feature. For example, higher-feature agents ought to be able to improve either a larger amount (i.e., have better posterior features given success), and/or improve at lower cost (i.e., $C_I$ should be negatively correlated with $x$).
3. Is it reasonable to assume that the decision-making policy $\pi$ is deterministic? Especially when discussing fairness, where pure classifiers often cannot achieve certain fairness guarantees, randomized policies might be able to do better.

---

> ### Author Response · Authors · 2024-09-23
> **Response to Reviewer 69Tp**
>
> Thanks for your comments. We address your concerns and revise the draft as follows.
>
> > Discussion of the assumptions
>
> - Agents will always choose to either manipulate or improve?
>
> Our model implicitly includes the action of "doing nothing". Specifically, our model assumes (1) improvement only succeeds with probability $q$; and (2) the improvement cost differs among agents which is modeled as a random variable $C_I$. For agents who do nothing, we may consider them as those who take improvement action at **zero costs** but **fail to improve**. Because no cost is incurred and the improvement fails, the resulting features remain the same and the outcome is equivalent to "doing nothing."
>
> Although it may be intuitive to model "doing nothing" as a separate action, we argue that it is more realistic in practice for individuals to always take an action. As pointed out by [Horowitz et al., 2024], agents who decide to participate in a system will pay an unavoidable cost. In the college admission example, all students applying to a college at least have finished all the requirements including high school curriculum, standardized tests, and application essays. This can be interpreted as "trying to improve". But the outcome of the improvement is non-deterministic as well as the cost.
>
> **We have added the above justification to footnote 3 and App. B.3 to make it clearer.**
>
> - Why aren't agents allowed to know their feature $x$?
>
> We agree with you that "students may know roughly how well they will do on an exam", so we assume the agents know their qualifications ($y$) and the conditional feature distributions ($P(x|y)$), instead of knowing their exact exam scores ($x$) before actually taking the exam.
>
> In the college admission example, the students may take some practice exams, where they can get get some knowledge of their score distributions from the practice exams. By comparing with the previous students who got admitted successfully (i.e., qualified individuals), individuals know whether they are already in qualified distribution, or need to cheat/study harder. However, the practice exam does not give students direct information on $x$ because the final exam score will be another random draw from $P_{X|Y}$ conditioned on $y$. Even if an agent is qualified and performs well in a practice exam, he/she can still do much worse in the final test although the random distribution of his/her final score differs from the unqualified ones. So it is unreasonable to directly infer the exact $x$ based on the practice exam. It only enables the student to know his $y$, and the final result is drawn from $P_{X|Y}$.
>
> We believe this setting is novel and more reasonable since students are neither a “prophet” (i.e., knowing the exact score they will get) nor a “fool” (i.e., knowing $P_X$ only without conditioning on $y$). In the whole framework, the best students can do is to estimate using conditional probabilities, and the hidden state encodes the information they have about themselves.
>
> - Improvement cost $C_I$
>
> Under our problem formulation, $C_I$ demonstrates the cost of acquiring qualification, i.e., improving from $y = 0$ to $y = 1$. Only individuals having $y = 0$ tend to improve, and their feature values $x$ are drawn from the conditional distribution $P_{X|Y}(x|0)$ which is a result of qualification $y$ and unknown to them. In other words, the cost is about improving to be qualified but not increasing $x$ values, which is a key difference between our model and the previous models.
>
> Meanwhile, there may be some sensitive attributes affecting $C_I$ for different social groups (e.g., socioeconomic status, race). Thus, our model permits different groups to have different $C_I$ and this will not influence the theoretical results on fairness.
>
> - Deterministic $\pi$
>
> We believe deterministic $\pi$ is practical, especially for high-stakes social domains where keeping the decision-making process transparent is crucial. Moreover, most previous works in classic strategic classification consider deterministic policies, so we believe it serves as a good starting point for strategic classification under the unforeseeable setting. However, we do appreciate your point of looking at randomized policies and this can be a meaningful future direction.

---

> > ### Author Response · Authors · 2024-09-23
> >
> > - Assumptions on continuity
> >
> > The assumptions on continuity on all probability density functions (PDF) are to ensure the cumulative distribution functions (CDF) are differentiable everywhere and closed form solutions exist for the non-strategic optimal policy without manipulation/improvement (Eqn. (2)). Note that the assumption is quite standard, we refer to the last paragraph on page 3 for more previous work using this assumption. Without the continuity, we need to discuss several special circumstances where PDFs only have finite supports, and the equilibrium analysis could be more challenging.
> >
> > - Why not assume $u = 1$?
> >
> > Your understanding is correct. It is WLOG to assume $u = 1$. Currently, we keep $u$ for better intepretability of the formula. We are happy to simplify the notation in the camera-ready version.
> >
> > - Robustness results
> >
> > In the experiment section, we present experiments to relax the assumption that the decision-maker knows $q, \epsilon$ exactly (Fig 5, 6). Reviewer xoU9 also endorses our work by saying "Robustness of the assumptions is also addressed in some of the experiments".
> >
> > > Numerical issues
> >
> > Thanks for pointing out the issues. **We have revised Figure 14,19,20 in the draft**.

---

> > > ### Comment · Reviewer_69Tp · 2024-09-23
> > >
> > > **Doing nothing**
> > >
> > > > For agents who do nothing, we may consider them as those who take improvement action at zero costs but fail to improve.
> > >
> > > In your model, this is only achievable by setting the improvement action to be equivalent to doing nothing. But that seems undesirable: is it not the whole point of the paper to model a case where both improvement and manipulation are possible?
> > >
> > > > Although it may be intuitive to model "doing nothing" as a separate action, we argue that it is more realistic in practice for individuals to always take an action. As pointed out by [Horowitz et al., 2024], agents who decide to participate in a system will pay an unavoidable cost. In the college admission example, all students applying to a college at least have finished all the requirements including high school curriculum, standardized tests, and application essays.
> > >
> > > This argument doesn't sufficiently justify this assumption in my mind. As a baseline, you are defining the non-strategic optimal policy as the optimal policy for the decision-maker assuming that everyone does nothing. Already here, you are implicitly assuming that it is reasonable to do nothing. The "unavoidable cost" would also be paid by a student who does nothing, so that doesn't change anything---the framing of the present paper suggests that $C_M, C_I$ should be interpreted as costs of the two actions *relative to* the cost of doing nothing.
> > >
> > > Again, concretely, my issue is this: if $C_M, C_I > 1$ then obviously the decision-maker should just play the non-strategic optimum $\hat\theta^*$ because no one will manipulate nor improve (because their cost for attempting either action is greater than the benefit they would derive). But this is not reflected in your model, because your model forces agents to not do nothing.
> > >
> > > **Improvement cost**
> > >
> > > > However, the practice exam does not give students direct information on $x$ because the final exam score will be another random draw from $P_{X|Y}$ conditioned on $y$.
> > >
> > > I guess my point is that students might be different *degrees* of unqualified: Student A might be far more qualified than Student B, even though they both have $y=0$. These students might learn where they are using practice exams: B will score quite poorly, whereas A might score close to the boundary. Thus the students can gain information about their $x$-value beyond what knowing $y$ alone would tell them.
> > >
> > > Perhaps both of these above points are things that would add complexity to the model beyond the scope of the present paper, but I would still at minimum like to see a bit more discussion here if possible.

---

> > > > ### Author Response · Authors · 2024-09-24
> > > >
> > > > Thanks for your insightful follow-up. We clarify your confusion as follows.
> > > >
> > > > > Further clarification on "doing nothing"
> > > >
> > > > - We claim "the action of "doing nothing" can be encoded as a special
> > > > case of improvement" from the perspective of the decision-maker (not the individuals). For individuals who decide to improve with randomized costs and randomized outcomes, it is possible that for certain individuals (those with zero improvement costs but fail to change the label), the realization of improvement outcomes can be equivalent to “doing nothing”. Thus, the decision-maker can regard these agents who “decide to improve (at zero costs) but not truly improve” as the ones who do nothing.
> > > >
> > > > - On the contrary and as we argued before, unqualified individuals in our model are not given an explicit option of "doing nothing" with a deterministic $0$ cost and $0$ utility in the first place. In our framework where improvement is modeled as the imitative behavior in social learning (the third paragraph on page 2), all unqualified individuals staying in the system are influenced by the desire to be qualified and must do something to imitate the qualified profiles because they know they are unqualified. **Otherwise, they will just quit the system and are not in our interest.** In the college admission example, all unqualified students who do not give up applications and do not cheat are doing similar things: striving to learn more courses, take the tests, and prepare for application packages. The only difference is the realization of the improvement outcomes (some students really improve, some do not). To concretely answer your question of $C_M, C_I$, we slightly modify the "budget" to "utility" in footnote 3: For all agents already in the decision-making system, they already have a sufficiently large initial utility exceeding $C_M, C_I$. This means if they choose to quit, they lose the initial utility.
> > > >
> > > > - Finally, the non-strategic policy as a baseline does not mean we endorse the plausibility of “agents can choose to do nothing”. Instead, the policy just represents traditional learning frameworks that assumed a static population distribution and did not account for individual strategic behavior (which indeed exist). In many previous works/simulations in strategic classification literature, all unqualified agents move their features in order to be admitted (e.g., the credit experiment in [Perdomo et al., 2021] and the ACSIncome-CA experiment in [Guldogan et al., 2022]).
> > > >
> > > >
> > > > > Improvement cost
> > > >
> > > > We understand your point of *different degrees of unqualified*. Currently, our model indeed assumes $y$ as the only additional information an individual has. While we agree that our framework might be more realistic if permitting $C_I, C_M$ to encode some information of $X$, this may lead to each individual having a different cost distribution and the corresponding equilibrium analysis can be extremely challenging. Mathematically, if we denote $K(x)$ as the information available for an inidvidual, then the manipulation/improvement costs and utilities must be written as $C_M(K(x)), C_I(K(x))$ and $U_M(\theta, K(x)), U_I(\theta, K(x))$. This directly results in $P_M(\theta) = E_{x}\{Pr[U_M(\theta, K(x)) > U_I(\theta, K(x))]\}$ to be an expectation over $X$. Analyzing the shape and monotonicity of $P_M$ analytically can be challenging, while empirical experiments may be possible. Meanwhile, theoretically characterizing $K(x)$ may also be a challenge since it is hard to quantify the "available information" in reality.
> > > >
> > > > We believe all these challenges are worthwhile to discuss and hope future work can cover them. We are also willing to add all the above discussion to the appendix.

---

### Review · Reviewer_FRV2 · 2024-09-05

**Summary Of Contributions:**

The paper introduces a new model to capture improvement and manipulation (in the context of strategic classification) based on a Stackelberg game between the decision maker and the participants. It provides a characterization of the best response policies, and decomposes the objective between non-strategic and strategic policies in terms of certain interpretable and natural terms that provide several insights. Based on this, the authors study how the preferences of the decision maker affects the optimal policy and its fairness properties. Finally, the theoretical results are validated through a series of experiments.

**Audience:**

Yes

**Broader Impact Concerns:**

The paper adequately addresses concerns regarding the ethical implications of the work.

**Claims And Evidence:**

Yes

**Requested Changes:**

I expect to see the following changes:

- A more clear explanation regarding the novelty of the model, clarifying which components were introduced in prior work and what is new.
- A more detailed discussion about some of the related work (see above).

I also have the following minor comments:

- Page 17: strategic manipulation means intervening -> Strategic manipulation means intervening
- The citations format is often used incorrectly: when using parenthesis, the citation should not be syntactically part of the sentence.

**Strengths And Weaknesses:**

Overall, the paper tackles a central problem in the interface of machine learning and game theory concerning strategic classification and naturally continues a long line of research on the subject; the topic is certainly of interest to the TMLR audience. In particular, the paper introduces a new model that is able to provide insights concerning important considerations in the context of actual applications, most notably the presence of unforeseeable outcomes; this differentiates the paper from much of the prior work. I also found the motivating examples the authors provide quite convincing (although it would be nice if the authors could include the examples in the appendix to the introduction, space limitations notwithstanding). The model is overall cleanly presented in a self-contained manner, without assuming familiarity with prior related papers. Moreover, the decomposition derived by the authors is quite informative, and cleanly conveys several insights about the problem, in particular concerning the impact of different preferences from the side of the decision maker, which the authors discuss in detail. All of the mathematical statements are informative and are discussed and explained in detail. A particularly interesting result is Theorem 4.6, which shows how fairness properties are affected by adjusting preferences. All statements appear to be sound, and I did not find any notable issues. The experiments are also quite thorough and support the theoretical findings.

In terms of weaknesses, the authors could do a better job of explaining the novelty of the introduced model in relation to prior work. That is, to explain clearly all aspects of the model that were introduced in prior work, and everything that is new. My understanding is that the model builds on, among others, Zhang et al. (2022), with some additions. I believe the authors can make this more clear in the revision. Besides this issue, one weakness is that all of the mathematical derivations are very straightforward, suggesting that the results are not too deep. Although the paper draws some interesting conclusions, one wonders whether those conclusions are fundamental or are just an artifact of the modeling assumptions. Finally, discussion regarding related literature is oftentimes lacking. For example, in the introduction it is claimed that prior work assumes that "individuals can perfectly foresee the outcomes of their behaviors when they best respond", but I believe that there are certain exceptions to that. For example, see Harris et al. ("Bayesian persuasion for algorithmic recourse"). Moreover, in the related work section in the appendix, I expected to see a more detailed discussion regarding certain related papers, in particular about the papers cited after "there are other studies considering both strategic manipulation and improvement" since they seem very much related.

---

> ### Author Response · Authors · 2024-09-23
> **Response to Reviewer FRV2**
>
> Thanks for your comments. We answer your questions and have revised the draft as follows:
>
> > Novelty
>
> On **page 2**, we further elaborate on how our models differ from [Zhang et al., 2022] (only considered manipulation) and [Liu et al., 2020] (only considered improvement) which are the most similar previous works. However, our work is the first architecture of a comprehensive probabilistic framework to model unforeseeable outcomes under **both manipulation and improvement**. This fills a significant gap in previous SC literature. More importantly, the theoretical results in section 4 reveal ways for the decision-maker to disincentivize manipulation, incentivize improvement and promote fairness simultaneously by adjusting preferences.
>
> > Related work
>
> We reorganize the related work section (App. A) by comparing more works considering randomness and both actions. We add **App. A.3 (page 16)** to discuss how our works differ from other works considering both actions. We include the comparison to [Harris et al., 2022] there. Specifically, this work mainly focused on the situation where the decision-maker can choose from disclosing partial information of the model (namely, persuasion) and the agents estimating the best response using the partial information. Since the randomness completely comes from the information revealed by the decision-maker, this setting is different from ours where the randomness comes from the unforeseeable and imitative nature of agent improvement and manipulation.
>
> We also correct the typos and the citations.

---

### Review · Reviewer_xoU9 · 2024-09-17

**Summary Of Contributions:**

The paper presents a variant of the strategic classification problem, formulated as a Stackelberg game between a decision maker (e.g., college admission) and a population of individuals (e.g., students). The novel piece in the formulation is that individuals's manipulation of their profiles may lead to unforeseeable outcomes. The authors decompose the difference between strategic and non-strategic decision-makers' utilities into three interpretable terms that describes the individuals qualification improvement (or the lack thereof) and manipulation. By further adjusting the decision policy function, the authors claim that the decisions can better improve fairness and disincentivize manipulation. Both theoretical analysis and empirical simulations are provided.

**Audience:**

Yes

**Broader Impact Concerns:**

The topic studied in this paper is related to the fairness in algorithmic decisions. I believe the discussion in this paper may contribute to better understanding of how to battle manipulation (e.g., cheating) in real world scenarios. No additional concerns noticed.

**Claims And Evidence:**

Yes

**Requested Changes:**

- To highlight the significance of the contribution, the authors should be more specific with what their main take-aways are. E.g., the current summary of contribution names "how adjusting .. preferences can affect ... fairness". If there is any real insights for practitioners here, they are worth explicitly describing upfront.
- While I like the decomposition of $\Phi$ into the three components and the interpretation of them, I believe it can be beneficial to explain further with a concrete motivating example what each terms mean. This can be important since few, if any, of the terms in the expression are directly observable; some sort of statistical estimation or model fitting is required. While this estimation aspect may seem tangential to the main purpose of the paper, it is central to any applications of the results. (If I understand correctly, in the FICO experiment, these quantities are again computed from certain distributions of the authors' choice, rather than estimated from data.)
- [Minor] Further proofreading can be useful (e.g., formatting issues with quotation marks).

**Strengths And Weaknesses:**

Strengths:
- The model extends the strategic classification problem by limiting the knowledge and action space of the individuals so that the benefit of manipulation is not observable a priori. This setting is often more realistic in practice and is well motivated.
- The content is overall well presented and readable, with clean notations.
- The theoretical results are complemented with empirical evidence from both simulated and real world data. Robustness of the assumptions is also addressed in some of the experiments.

Weaknesses:
- I find the results overall quite straightforward. Given the similarity to the existing setting of strategic classification in the literature, I do not find the formulation truly novel. The theoretical results are standard once the assumptions are established; in particular, several results (e.g., Thm 4.1, Prop 4.2) are existential and non-quantitative.
- I am not sure how significant the results are regarding real world problems, as the setting seems overly simplifying (specifically, regarding the underlying distribution and the knowledge of type, but not the feature).

---

> ### Author Response · Authors · 2024-09-23
> **Response to Reviewer xoU9**
>
> Thanks for your comments. We answer your questions and have revised the draft as follows:
>
> > Significance of the contribution
>
> - Our work is the first architecture of a comprehensive probabilistic framework to model unforeseeable outcomes under both manipulation and improvement. This fills a significant gap in previous SC literature. More importantly, the theoretical results in section 4 reveal ways for the decision-maker to  disincentivize manipulation, incentivize improvement and promote fairness simultaneously by adjusting preferences. We believe these results are insightful for socially responsible decision-making under strategic settings.
>
> - We have revised the draft by highlighting the **real insight for practitioners** by by adding a concrete guideline for practitioners to apply our model at the end of section 4 **(page 9)**. We also emphasize that our theoretical results can be used in practice in abstract and the last paragraph **(point 3)** in introduction.
>
> > Example of the terms in the decomposition
>
> We add an example on college admission for the practical meanings of $\phi_1, \phi_2, \phi_3$ in the decomposition on **page 6**.
>
> > Statistical estimation/model fitting
>
> Our guideline practitioners to apply our model at the end of section 4 **(page 9)** includes discussions of the estimation. We provide an end-to-end guide to estimate the model parameters in practice in **App. B.5 (page 19)**. With this estimation procedure, practitioners can estimate the model parameters from data with controlled experiments.

---

### Decision · Action_Editor_yqvq · 2024-10-23

**Recommendation:** Accept as is

**Comment:**

The paper tackles the problem of machine learning under strategic behavior of participants who may manipulate their features to try and game the system. The paper relaxes common assumptions in prior literature on this topic with a proposed model where participants cannot easily guess the outcomes of counterfactual features, and instead, imitate other participants with positive labels as a proxy. This model and associated results are certainly of interest to a non-trivial subset of TMLR's audience. The reviewers have provided thorough reviews and are convinced of the correctness of the results, thereby meeting TMLR's criteria.

**Audience:**

Yes

**Claims And Evidence:**

Yes, the claims made in the submission are supported by clear and convincing evidence.